# Brief Histories of Retroviral Integration Research and Associated International Conferences

**DOI:** 10.3390/v16040604

**Published:** 2024-04-13

**Authors:** Duane P. Grandgenett, Alan N. Engelman

**Affiliations:** 1Department of Molecular Microbiology and Immunology, School of Medicine, Saint Louis University, St. Louis, MO 63104, USA; 2Department of Cancer Immunology and Virology, Dana-Farber Cancer Institute, Boston, MA 02215, USA; 3Department of Medicine, Harvard Medical School, Boston, MA 02115, USA

**Keywords:** integrase, retroviral integration, provirus, integrase strand transfer inhibitor, raltegravir, elvitegravir, dolutegravir, bictegravir, cabotegravir, allosteric integrase inhibitor

## Abstract

The field of retroviral integration research has a long history that started with the provirus hypothesis and subsequent discoveries of the retroviral reverse transcriptase and integrase enzymes. Because both enzymes are essential for retroviral replication, they became valued targets in the effort to discover effective compounds to inhibit HIV-1 replication. In 2007, the first integrase strand transfer inhibitor was licensed for clinical use, and subsequently approved second-generation integrase inhibitors are now commonly co-formulated with reverse transcriptase inhibitors to treat people living with HIV. International meetings specifically focused on integrase and retroviral integration research first convened in 1995, and this paper is part of the *Viruses* Special Issue on the 7th International Conference on Retroviral Integration, which was held in Boulder Colorado in the summer of 2023. Herein, we overview key historical developments in the field, especially as they pertain to the development of the strand transfer inhibitor drug class. Starting from the mid-1990s, research advancements are presented through the lens of the international conferences. Our overview highlights the impact that regularly scheduled, subject-specific international meetings can have on community-building and, as a result, on field-specific collaborations and scientific advancements.

## 1. Introduction

Retroviral integration research, with a history spanning 60 years, is deeply rooted in the traditional biological disciplines of genetics, virology, biochemistry, structural biology, and cell biology. The first international conference on retroviral integrase (IN) was held in 1995, and the most recent meeting—The 7th International Conference on Retroviral Integration (a.k.a. Retrointegration2023)—occurred between 31 July and 4 August 2023, in Boulder, Colorado. This treatise forms part of the *Viruses* Special Issue on papers presented at the 7th International Conference. Herein, we highlight early major research accomplishments in the retroviral IN and integration fields that occurred prior to the initial 1995 meeting. Starting from circa 1995, we then connect field advancements with presentations made at these historical conferences. Although research presented at the 2023 Boulder meeting will be presented in the Editors’ overview of the Special Issue and is accordingly not described in detail herein, our paper builds upon Dr. Grandgenett’s historical perspective of the IN/integration fields that was given as the closing oral presentation at the recent meeting.

To follow the details, it is important to appreciate the key steps and players in the process of retroviral integration. To start, reverse transcription of retroviral genomic RNA yields linear double-stranded DNA containing a copy of the viral long terminal repeat (LTR) at each end. Each LTR is composed of U3RU5 sequences, so the upstream terminus of retroviral DNA is composed of U3 sequences and the downstream end is made of U5 sequences. Retroviral INs possess two catalytic activities central to the integration of viral DNA into host cell chromosomal DNA. In a reaction referred to as 3′ processing, IN hydrolyzes the U3 and U5 LTR ends adjacent to invariant CA-3′ dinucleotides, resulting in chemically reactive CA_OH_-3′ ends [1,2]. During the ensuing strand transfer reaction [3,4], IN uses the 3′-oxygen atoms to make a staggered, double-stranded cut in chromosomal target DNA, which joins the viral 3′ ends to the host DNA 5′ phosphates [5,6]. The resulting recombination intermediate, with unjoined viral 5′-ends, is repaired by cellular machinery to yield the integrated provirus flanked by a short target site duplication (TSD) of the staggered DNA cut sequence (Figure 1). Depending on the viral species, the size of retroviral IN-mediated TSDs vary from 4 bp to 6 bp.

In their simplest forms, IN 3′ processing and DNA strand transfer reactions are staged using double-stranded oligonucleotides that model the terminal ~20–25 bp of viral U3 or U5 ends. Under these conditions, most strand transfer reaction products result from the integration of one LTR end into a second LTR end, the second molecule in this case playing the role of host chromosomal DNA. Integration during virus infection, by contrast, requires IN action on both U3 and U5 LTR ends, which is referred to as concerted integration activity. Modified reaction conditions that monitored the integration of two LTR ends, as well as the ability for these strand transfer events to generate the expected TSD, were required to study retroviral IN concerted integration activities [3,7,8,9,10] (Figure 2). Structural characterizations of retroviral intasomes in more recent years have elucidated the architectural details of how multimers of IN bind and synapse two ends of linear viral DNA together to catalyze 3′ processing and strand transfer of the DNA ends in concerted fashion (see ref. [11] for a recent review).

An important consequence of basic scientific research is the ability to inform the development of medicines for the betterment of society, and the field of HIV-1 IN research has witnessed superlative translational successes. DNA oligonucleotides that mimicked the products of IN 3′ processing activity importantly supported strand transfer activity in vitro [3,4,12] (Figure 2). Despite some initial speedbumps whereby suboptimal assay designs led to identification of off-target compounds or clinically nonviable leads [15,16], pre-binding HIV-1 IN to such pre-cleaved strand transfer reaction substrates led to the identification of diketoacid (DKA)-based compounds that preferentially inhibited IN strand transfer activity [17,18]. With the eventual development and rollout of raltegravir (RAL) in 2007, IN strand transfer inhibitors (INSTIs) made an immediate clinical impact, and the second-generation inhibitors dolutegravir (DTG) and bictegravir (BIC) impart much higher barriers to the generation of drug resistance than do the predecessor compounds RAL and elvitegravir (EVG) (reviewed in ref. [11]). Most commonly co-formulated with nucleoside reverse transcriptase (RT) inhibitors (NRTIs), second-generation INSTIs are now recommended for drug-naïve patients as well as for people living with HIV (PLWH) who have not previously failed an INSTI-containing regimen [19]. Recently, an injectable long-acting INSTI (cabotegravir; CAB) has been shown to significantly reduce the risk of HIV-1 acquisition [20] (Table 1).

## 2. Early Studies

Early advancements in retroviral research leveraged animal viruses that were discovered via their abilities to cause disease in susceptible hosts. The virus family *Retroviridae* includes two subfamilies, Orthoretrovirinae and Spumavirinae. Orthoretroviruses that served key roles in the early days of retroviral integration research include cancer-causing α-retroviruses of birds, such as AMV, and leukemogenic γ-retroviruses of mice, such as Mo-MLV. Although it is non-pathogenic in animal hosts, the Simiispumavirus PFV would later rise to prominence due to its value as a structural biology model for IN-viral DNA complexes (a.k.a. intasomes) [11,21].

Retroviral integration research can be traced to Howard Temin’s brainchild that the α-retrovirus Rous sarcoma virus (RSV) must replicate through a proviral DNA intermediate [22]. Because RSV particles were known to harbor RNA (and not DNA), this suggestion at the time was practically heretical and Temin, needless to say, received significant pushback from colleagues. However, within a relatively short timeframe, both he and David Baltimore provided key evidence as to how retroviruses could replicate through a DNA intermediate: RSV virions [23], as well as Rauscher murine leukemia virus [24] particles, harbored RNA-dependent DNA polymerase activity, and the presumptive enzyme was dubbed “reverse transcriptase” by a Nature editor. Building off this central tenet that retroviral particles harbor within them enzymes needed to complete essential early replication steps, one of us (Grandgenett) formulated the following two-part hypothesis that led to the discovery of retroviral IN: (1) the protein must be present in purified core particles along with RT and viral RNA and (2) it must bind DNA and should possess DNA endonuclease activity for integration of the viral DNA into the host DNA genome. These properties were associated with a 32 kDa protein purified from isolated AMV cores using RNA-affinity chromatography during the purification process [25]. The purified protein was capable of nicking supercoiled DNA in the presence of Mg^2+^, and it bound DNA in a nitrocellulose DNA-binding filter assay. The protein was proteolytically derived from one of the β subunits (96 kDa) of the precursor RT ββ homodimer, which results in the active αβ heterodimeric form of RT found in α-retroviruses [26]. NH_2_- and COOH-terminal amino acid sequence analyses of the purified dimeric IN and RT αβ subunits helped to establish that partially phosphorylated IN was derived from the carboxyl terminus of the β subunit [27,28,29,30,31]. The NH_2_-terminus of HIV-1 IN (32 kDa) was subsequently characterized in 1986 [32], thus allowing the cloning and expression of the recombinant protein by many laboratories.

Genetic analysis of different retrovirus species revealed that a function provided by the 3′ end of their *pol* genes was required for integration of viral DNA. A single missense mutation or short deletion of Mo-MLV [33,34] and a short internal deletion of spleen necrosis virus (SNV) [35] established that these regions supplied a *trans*-acting factor required for integration. Viral DNA synthesis was not obviously affected by these *pol* gene changes. Two missense mutations near the NH_2_-terminus of α-retrovirus IN also failed to affect viral DNA synthesis, or for that matter integration, affecting only proteolytic processing of the β RT subunit [28,36]. Genetic approaches also helped to determine the extent of LTR sequences required for integration [37,38] and to determine that IN was responsible for 3′ processing of Mo-MLV DNA ends during virus infection [39]. Subsequent work that mainly focused on HIV-1 revealed that numerous IN mutations caused pleiotropic replication defects due to disruption of viral core formation and, as a consequence, led to DNA synthesis defects during the subsequent round of virus infection [40,41]. Such mutations have been labelled “class II” to distinguish them from class I IN mutations that primarily impact the integration step of the viral lifecycle [42]. Some mutations in Mo-MLV IN also yield reverse transcription defects [43]. A comprehensive analysis is required to ascertain the universality of class I versus class II IN mutant phenotypes across divergent retroviral species. 

One key unknown in the early 1980s was the structure of the precursor form of retroviral DNA for integration. In addition to linear viral DNA, reverse transcription yields several circular forms, including 1- and 2-LTR containing circles. At the time, knowledge of DNA recombination mechanisms was largely confined to bacterial systems. DNA transposition, which resulted in the movement of a DNA transposon from one location to any number of new locations, yielded a short TSD flanking the newly transposed element. Integration of bacteriophage lambda, which by contrast occurred at a specific site in the bacterial chromosome, proceeded via a circular phage DNA precursor and did not generate any additional DNA sequences. The sequencing of proviral-host junctions by several labs in 1980 revealed short repeat sequences flanking the integrated viruses, indicative of transpositional recombination [44,45,46,47]. By introducing the LTR-LTR circle junction into an interior region of SNV, a now-infamous 1984 paper from Panganiban and Temin claimed that integration could occur via the 2-LTR circle junction [48]. This model, which borrowed from bacteriophage lambda integration, seemed valid because retroviral circle junctions harbor twofold symmetrical sequences, as did the bacteriophage attachment site, and subsequent biochemical efforts with purified IN proteins accordingly focused on circular LTR substrate DNAs [49]. Uncertainty remained until the publication of a breakthrough paper from Patrick Brown and colleagues in 1987. These investigators showed that extracts of cells infected with Mo-MLV possessed the ability to integrate the viral DNA made during the cell infection into an exogenously added target DNA in vitro. Careful analysis of the integration activities of cytoplasmic versus nuclear extracts proved that linear viral DNA was active for integration, casting doubt on the 2-LTR circle model [50]. Integration activity, moreover, tracked with a virus-derived high molecular weight complex termed the “pre-integration complex” or PIC for short [51,52]. One drawback of the 1987 study was that it scored integration via a genetic assay that required complementation of a suppressor bacterial strain following phage-mediated transduction of the in vitro integration product. Soon after, Fujiwara and Mizuuchi greatly simplified the assay by using Southern blotting to detect the products of integration. They additionally probed the structure of the unjoined viral DNA 5′ ends in the integration intermediate, which were 2 base extensions as compared to the 4-base extensions predicted for integration via the 2-LTR circle junction [5]. These breakthrough studies refocused biochemical efforts on linear LTR substrate DNAs, which in relatively short order defined bona fide IN 3′ processing and DNA strand transfer reaction conditions [1,2,3,4]. 

The simplicity of the in vitro 3′ processing and DNA strand transfer reactions led to a flurry of biochemical studies in the early 1990s. Although these studies are too numerous to comment on in much detail, highlights included elucidation of LTR bases required for IN activity [53,54,55] and discovery of a reversal of strand transfer activity called “disintegration” [56] that had less stringent requirements than strand transfer and would thus become important for characterizing mutant IN enzymes [57]. Parallel work defined IN as a 3-domain protein composed of the N-terminal domain (NTD), catalytic core domain (CCD), and C-terminal domain (CTD) [57,58,59,60] and characterized the roles of conserved amino acid residues (Figure 3). The NTD harbors HHCC amino acids that are conserved across retroviral and retrotransposon INs and that leverage zinc binding to fold into a helix-turn-helix [57,61]. The CCD harbors the IN active site, which is composed of electronegative Asp and Glu residues [59,62,63] and is conserved as a DDE triad among the mobilizing enzymes of an expanded superfamily [64]. The DDE carboxylate side chains coordinate the positions of two Mg^2+^ ions that enhance the nucleophilicity of attacking oxygen atoms (water for 3′ processing; 3′-oxygens of viral DNA for strand transfer) and destabilize associated scissile phosphodiester bonds [65].

## 3. International Conferences on Retroviral Integration

We will now focus on the six international conferences held from 1995 to 2017. We will frame presentations given at the conferences within the context of subject-specific sessions that were utilized to organize the meetings. Due to space limitations, we were unable to comment on all 214 talks and instead selected six to nine talks per conference to provide a general overview of field advancements and/or talks that were particularly relevant to the development of the clinical inhibitors (Table 1). We apologize upfront to any speaker who thinks we were remiss in omitting their talk from this discussion. To enhance inclusivity, we supply meeting-specific tables (Table 2, Table 3, Table 4, Table 5, Table 6 and Table 7) that list all speakers and, when known, talk titles. The following summaries were chosen solely from oral presentations (and not from poster presentations).

### 3.1. The First Retroviral IN Conference

The first international conference was held on 19–20 January 1995 at the National Institutes of Health in Bethesda, Maryland, USA. The title was “NIH Conference on Retroviral Integrase—Molecular Biology and Pharmacology—A Novel Target for the Treatment of AIDS” (Table 2). The organizers were Drs. Robert Craigie and Yves Pommier. The number of registrants was 158. 

**Table 2 viruses-16-00604-t002:** Talks presented at the 1995 Bethesda conference ^1^.

Session	Speaker	Title
I	Harold E. Varmus	Retroviral integration
	Kiyoshi Mizuuchi	DNA transposition and retroviral integration: Similarities and differences
	John M. Coffin	Specificity of viral DNA integration in vitro and in vivo *
	Stephen P. Goff	A human homologue of the yeast Snf5 transcription factor binds and stimulates HIV-1 integrase
	Frederic D. Bushman	HIV-1 preintegration complexes *
	Patrick O. Brown	Genetic footprinting: Using integrase as a tool to study gene function
II	Anna Marie Skalka	Molecular mechanisms in retroviral integration
	Abhijit Mazumder	Processing of modified DNA substrates by HIV-1 integrase
	Monica J. Roth	Coordinated disintegration reactions of M-MuLV integrase
	Duane Grandgenett	Efficient concerted integration of retrovirus-like DNA by integrase from avian myeloblastosis virus and HIV-1 *
	Tamio Fujiwara	Analysis of in vitro HIV-1 DNA integration reaction by UV cross-linking
	Samson A. Chow	In vitro activities of HIV-1 integrase/*E. coli* LexA fusion proteins
III	Robert Craigie	Improving the solubility properties of HIV-1 integrase *
	Frederick Dyda	Three dimensional structure of the core domain of HIV-1 integrase *
	Yves Pommier	Structure-activity relationships of drugs that inhibit HIV-1 integrase
	Richard A. Houghten	The future possibilities and current realities of soluble low molecular weight synthetic combinatorial libraries: A revolution in basic research and drug design
	Ronald H. A. Plasterk	The integrase proteins of HIV-1 and HIV-2; Potential targets for anti-HIV drugs
	Jean-Francois Mayaux	High-throughput screening for HIV-1 integrase inhibitors: One year after
	Jean-François Mouscadet	Triplex-mediated inhibition of HIV DNA integration in vitro
	Daria J. Hazuda	The use of immobilized substrates to identify and characterize inhibitors of HIV-1 integrase *

^1^ Talks highlighted in main text denoted by *.

#### 3.1.1. Session I: Retroviral Integration In Vivo

HIV-1 PIC characterizations and DNA target site selection by α-retroviral IN were emphasized in the initial meeting session. Work presented by Frederic Bushman established that small molecule inhibitors previously claimed to inhibit purified HIV-1 IN activity in vitro were much less able, or unable, to inhibit PIC-mediated strand transfer activity [15]. These data suggested that purified IN was not necessarily a reliable identifier of clinically relevant inhibitors, which, as mentioned in the Introduction, was mainly due to the types of assays that had been used up until this point in time to identify potential inhibitors. Dr. Bushman further elucidated detailed structure/function characterizations of partially purified HIV-1 PICs. These were large, high-molecular weight complexes with an estimated Stokes radius of 28 nm that contained at least 100 molecules of IN along with the viral matrix protein, RT, and capsid protein (CA) [67]. PICs contained both blunt-ended and 3′-processed viral DNA ends, with only the recessed ends being active for integration as determined by cell-based kinetic studies of viral DNA synthesis and LTR end processing. The viral DNA ends were linked together by a protein bridge that protected the ends from external nuclease digestion, which we infer today was likely an early mapping of the HIV-1 intasome. The DNA plus-strand was found to contain several gaps, and the gapped DNA was competent for integration in vitro [68]. These studies further expanded our knowledge of HIV-1 PICs, which were critical reagents that helped to authenticate the first clinically relevant IN inhibitors [18].

In addition to γ-retroviruses [69] and retrotransposable elements in yeast [70], α-retroviruses provided a tractable model to understand the molecular mechanisms of DNA target-site selection in vitro and in virus-infected cells. John Coffin described the development of a locus-specific PCR/primer-extension method to define cellular regions and sequence preferences for insertion of α-retroviral DNA in cells, as well as PCR-based methods to detect integration into plasmid DNAs in vitro using either cell-derived PICs or recombinant protein as the source of IN activity [71,72]. Integration patterns in vitro were distributed across the examined regions of target DNA, were highly non-random, and were more related to local DNA structure than to sequence. Methylation of C nucleotides, moreover, created strong target-site preferences [72]. PICs and recombinant IN yielded overall similar patterns, indicating the results were largely driven via IN-LTR DNA interactions with target DNA. In cells, most integration sites were widely distributed across the avian genome, with apparently little to no regional avoidance of specific sequences. The PCR-based method to identify integration sites in cells was a forerunner of future ligation-mediated (LM)-PCR techniques, which, when combined with the sequencing of whole cell genomes, mapped retroviral integration sites at single-nucleotide resolution and at scale [73,74,75].

#### 3.1.2. Session II: Molecular Mechanisms in Retroviral DNA Integration

It was at the time—and still remains—difficult to isolate PICs from virus-infected cells in the quantities required for extensive purification and highly detailed characterizations. The next best thing was accordingly to determine reaction conditions for efficient concerted integration activity, assays that eventually became critical to define functional intasome complexes for structural biology. Duane Grandgenett reported efficient concerted integration of a linear virus-like DNA donor (487 bp), with U5 and U3 LTR ends pre-processed via restriction endonuclease digestion, into a supercoiled plasmid target DNA using IN purified from AMV cores [7]. In an experiment employing donor DNA 5′-end labeled at the noncatalytic LTR strand for quantitative purposes under optimized assay conditions, approximately 50% of the donor/target recombinants were determined to result from the concerted integration of both donor DNA ends. The insertion of one LTR end into the target DNA, referred to as half-site integration (Figure 2), was the other major product. Adapting these seminal findings to the HIV-1 system using similarly sized LTR donor DNA and nonionic detergent-lysed virions as a source of IN activity produced 5–10% concerted integration products [76]. These studies, importantly, advanced our understanding of the reaction conditions required to analyze the concerted integration activities of different virus particle-derived IN proteins and suggested the possibility to screen for specific HIV-1 IN inhibitors based on extracts of nonionic detergent-lysed virions.

#### 3.1.3. Session III: Structural Studies and Initial Efforts to Identify HIV-1 IN-Specific Inhibitors

Full-length HIV-1 IN (288 amino acid residues) purified in recombinant form possessed comparatively poor solubility properties, which greatly limited reasonable chances for successfully solving its 3-dimensional structure given the X-ray crystallographic methodologies popular at the time. To circumvent this shortfall, investigators took to expressing the various IN domains on their own. Although it was insoluble when expressed in bacteria, Robert Craigie found that the HIV-1 IN CCD could be extracted under denaturing conditions (8 M guanidine-HCl) and, following column chromatography, effectively refolded in the presence of CHAPS detergent [57,77]. Although defective for 3′ processing and strand transfer activities, the refolded CCD, importantly, supported disintegration activity, revealing that it possessed a functional IN active site. Despite this important finding, such CCD preps remained refractory to crystallization. In a landmark study, Dr. Craigie described a systematic approach to counteract the inherent insolubility of the active site domain protein by site-specific modification of hydrophobic amino acid residues. While isolated hydrophobic residues were replaced by lysine, adjacent residues were replaced by the same number of alanines. In all, 29 different missense mutants of the CCD were constructed and characterized [78]. A lysine substitution for phenylalanine at position 185 (F185K) dramatically increased the solubility of the CCD to 25 mg/mL, and biophysical measurements revealed the CCD^F185K^ to be a monodispersed dimer. These studies laid the groundwork for the first structural characterization of a retroviral IN domain protein.

Fred Dyda described the crystallization and structural determination of CCD^F185K^ to 2.5 Å resolution [79]. The globular structure contained a five-stranded β sheet flanked by helical regions, and the overall topology was highly similar to previously solved polynucleotidyl transferase enzymes RNase H and recombination UV (Ruv) C. The IN active site was identified by the position of the two aspartic acid residues of the DDE motif; the glutamic acid was disordered due to its positioning on a flexible element whose structural solution would eventually benefit from the inclusion of viral substrate DNA. CCD^F185K^ formed a dimer with an extensive dimeric interface. Because HIV-1 integration generates a 5 bp TSD, the two scissile phosphodiester bonds in B form target DNA would be separated by ~17 Å. The two active sites in the CCD^F185K^ dimer were, however, separated by 35 Å, implying that the active oligomeric form of IN for strand transfer was at least a tetramer. These results were quickly followed by the publication of the X-ray structure of the α-retroviral CCD by Wlodawer and colleagues [80]. 

The in-solution integration reactions catalyzed by virion-derived IN, recombinant IN, and PICs were transformative biochemical tools. However, a simplified, scalable assay was required to screen thousands of compounds as potential HIV-1 IN inhibitors. Daria Hazuda described a non-radioisotopic microtiter plate assay for HIV-1 IN strand transfer activity that leveraged immobilized LTR substrates and biotinylated target DNA [81]. The system was used to screen potential inhibitors and study interactions between IN with LTR substrate and target DNAs. The IN-LTR complex, using donor DNA containing either blunt-ended or 3′-recessed ends, was stable in the absence of target DNA. Small-molecular-weight inhibitors of IN that inhibited strand transfer when they were added at the time of integration complex assembly with target DNA were identified. This microtiter plate assay system importantly established conditions with pre-cleaved LTR DNA that eventually led to the identification of preclinical DKA INSTIs [17,18]. 

### 3.2. The Second Retroviral IN Conference

The second conference was held on 28–30 October 2001, in Paris, France. The title was “2^nd^ International Conference on Retroviral Integrase—A novel target for the treatment of AIDS”. The organizers were Jean-Francois Mousçadet, Jean-Claude Brochon, and Yves Pommier, who, together with Christian Auclair, Frederic Bushman, Zeger Debyser, and Anna Marie Skalka, served as the scientific committee. There were 109 registrants. 

Starting with this meeting, the conferences began with a keynote talk. The 2001 keynote talk, given by Erik De Clerq, was entitled “Introduction to anti-HIV drugs and targets” (Table 3).

**Table 3 viruses-16-00604-t003:** Talks presented at the 2001 Paris conference ^1^.

Session	Speaker	Title
Keynote	Erik De Clerq	Introduction to anti-HIV drugs and targets
I	Alexandre Wlodawer	Avian sarcoma virus integrase as a model for detailed studies of retroviral integrases *
	Robert Craigie	Structure of a two-domain fragment of HIV-1 integrase: implications for domain organization *
	Jean-Claude Brochon	Characterization of oligomeric state of HIV-1 integrase and oligonucleotide-integrase complex in vitro by dynamic fluorescence
	Robert Stroud	What does the two-domain structure of HIV-1 integrase tell us about its function? *
	Michael Katzman	The nonspecific nuclease activity of retroviral integrase: Utility and potential
	Allison A. Johnson	The effect of DNA modifications on catalysis by HIV-1 integrase
	Emmanuel A. Faust	Stimulation of human flap endonuclease-1 by human immunodeficiency virus type 1 integrase
II	Stephen P. Goff	Role of the p12 Gag protein of Moloney murine leukemia virus in early events of infection *
	Duane Grandgenett	Efficient concerted integration by recombinant HIV-1 integrase without cellular or viral factors
	Zeger Debyser	Cell biology of HIV-1 integrase overexpressed in human cells *
	Catherine Dargemont	Mechanisms for nuclear import of HIV-1 integrase
	Frederic D. Bushman	Structure and function of integration complexes
	Bénédicte Van Maele	Kinetics of HIV-1 integration as analyzed by quantitative PCR
III	John M. Coffin	Specificity of retroviral integration
	Alan Engelman	Nuclear localization of HIV-1 preintegration complexes
	Rene Daniel	Role of host repair function
	Jonathan Leis	Changes in the mechanism of concerted integration induced by base substitutions in the HIV-1 U5 and U3 terminal sequences in vitro
	Richard S. Kornbluth	Cellular activation is required to generate the host cell factors needed for the formation of integration competent HIV-1 preintegration complexes in primary T cells
	Marie-Claude Lang	Do retroviruses preferentially integrate within highly plastic regions of the human genome?
IV	David R. Davies	The binding of inhibitors to the core domain of HIV-1 integrase *
	Simon Litvak	Searching HIV-1 integrase ligands, finding putative protein cofactors and specific inhibitors
	Yves Pommier	From nucleic acids to diketo acids *
	Jean-François Mouscadet	Mechanism of action of styryl-quinolines as anti-integrase reagents

^1^ Talks highlighted in main text denoted by *.

#### 3.2.1. Session I: Crystal Structures and Molecular Mechanisms of HIV-1 and α-Retroviral Integration

Structural characterizations of isolated HIV-1/2 NTDs and CTDs [82,83,84,85], as well as α-retroviral CCD constructs, had expanded greatly since the initial 1995 conference. Alex Wlodawer presented X-ray crystal structures of wildtype (WT) and active site D64N mutant α-retroviral CCD proteins and compared these structures with those of previously solved polynucleotidyl recombinases such as the HIV-1 CCD, RNase H, and Ruv C [80,86]. These studies, importantly, kickstarted X-ray crystallography approaches using protein substrates other than HIV-1 IN across multiple laboratories, which culminated several years later with the first structure of a retroviral intasome [21].

Important new progress was presented on the two-domain structures of HIV-1 IN. Robert Craigie elucidated the spatial arrangement of the NTD-CCD two-domain fragment (residues 1-212) containing solubility-enhancing changes W131D, F139D, and F185K [87]. The structure revealed a dimer interface between NTDs that differed from that observed previously for the isolated NTD [84]. Superposition of the NTD-CCD structure with the CCD-CTD two-domain structure reported by Stroud and colleagues in 2000 [88] (see below) resulted in a plausible model for the full-length HIV-1 IN dimer. Further analysis revealed structural resemblance to the Tn5 bacterial transposase and solvent-accessible channels that were suitable for DNA binding. 

Robert Stroud reported the first structure of a multidomain HIV-1 IN fragment, the CCD-CTD (amino acid residues 52-288) with solubility-enhancing C56S/W131D/F139D/F185K/C280S changes [88]. The structure, which was resolved at 2.8 Å, revealed a Y-shaped dimer. Within the dimer structure, the CCDs formed their canonical interface, while the CTDs were located 55 Å apart from one another. The structure of the CCD fragment (residues 52-210) with C56S/W131D/F139D/F185K changes, which was resolved at 1.6 Å, was identical to the CCD within the two-domain construct. These above structural studies significantly advanced our understanding of the overall structure of HIV-1 IN and how the different protein domains might interact with one another within the active multimeric complex. However, future efforts with active intasomes were surely necessary to elucidate the molecular details of the IN-IN and IN-viral DNA interactions important for concerted integration of HIV-1 LTR ends [89,90,91]. 

#### 3.2.2. Session II: PICs and IN Cell Biology

Although it was known to contain elements important for virus assembly, Steve Goff presented the unexpected finding that the p12 product of the Mo-MLV Gag polyprotein also played a key role in completing the early events of the viral lifecycle. Viruses mutated for p12 produced normal levels of reverse transcripts, and the 3′ ends of these viral DNAs were, moreover, processed normally by IN. However, some of the mutant viruses were defective for formation of nuclear forms of virus DNA, including 2-LTR circles and integrated proviruses [92,93]. These studies spearheaded the role of retroviral Gag-derived proteins in enabling PICs to access the nuclear environment and host cell chromatin for integration, a finding that would in future years extend to the spumaretrovirus Gag protein [94] and HIV-1 CA [95,96]. As structurally determined for PFV Gag [97], Mo-MLV p12 seems certain to employ a conserved Arg residue to anchor the PIC to the acidic patch of the histone 2A/2B heterodimer [98].

Goff’s lab previously identified the first cellular factor to bind HIV-1 integrase, IN-interactor 1 (INI1), which is the SMARCB1 component of the large SWI/SNF chromatin remodeling complex [99] (Table 2). Although INI1 could stimulate HIV-1 IN strand transfer activity in vitro, subsequent work indicated that it primarily works post-integration to regulate the functionality of HIV-1 virus particles [100]. There was, accordingly, significant interest in identifying novel cellular cofactors that may associate with HIV-1 PICs to regulate early replication events, including integration into the cellular genome. As a novel approach to cellular cofactor identification, Zeger Debyser described optimization of HIV-1 IN expression in recombinant form in human cells, which in part leveraged codon-optimization to overcome the inherent instability of the protein under these expression conditions [101]. Stable expression of the IN was achieved in different cell lines, including HEK293T and CEM T cells. IN at steady-state was exclusively nuclear, was associated with cellular DNA and nuclear proteins, and was extracted from cells as a stable tetramer. Importantly, this IN was active during integration, as shown by functional trans-complementation of incoming integration-defective HIV-1-derived vector particles. Within 2 years, Debyser and colleagues would leverage these cells to identify LEDGF/p75 as a dominant HIV-1 IN cellular binding factor [102].

#### 3.2.3. Session IV: HIV-1 IN Inhibitors

Although Hazuda and colleagues had recently published their landmark DKA inhibitor paper [18], the rollout of the first clinical INSTI was still several years away. Thus, orthogonal approaches to HIV-1 IN inhibitor identification and characterization remained quite topical at the time. David Davies described the co-crystal structure of the small molecule inhibitor 5CITEP with the HIV-1 CCD, which was resolved to 2.1 Å resolution [103]. The inhibitor bound centrally to the IN active site, and in a multiwell plate-based integration assay inhibited 50% of HIV-1 IN strand transfer activity at 2.1 µM. Although 5CITEP at the time was a reasonable lead compound, its pre-clinical development would be supplanted by the DKA compounds. 

Yves Pommier compared the mechanisms of action of 5CITEP and the Merck DKA L-708,906 [104]. 5CITEP inhibited IN 3′ processing activity at concentrations whereby L-708,906 solely inhibited IN strand transfer activity. Dr. Pommier also described a novel bifunctional DKA compound that inhibited 3′ processing more potently than 5CITEP. Using LTR substrates with defined modifications, the bifunctional DKA was determined to bind both acceptor (target DNA binding) and donor (LTR binding) sites on HIV-1 IN, while L-708,906 bound selectivity to the acceptor site. Indeed, Merck independently described the DKAs as competitors of target DNA binding to IN [105].

### 3.3. The Third Retroviral IN Conference

The third international conference, titled “3^rd^ International Conference on Retroviral Integrase—Molecular Biology and Pharmacology”, was held 14–18 September 2008 at the Marine Biological Laboratory in Woods Hole, Massachusetts, USA. The organizers were Alan Engelman, Robert Craigie, and Yves Pommier. There were 84 registrants and Frederic Bushman presented the keynote talk. While most of the other talks were given by invited speakers, three short talks, chosen from amidst the submitted abstracts by the meeting organizers, were also presented (Table 4). 

**Table 4 viruses-16-00604-t004:** Talks presented at the 2008 Woods Hole meeting ^1^.

Session	Speaker	Title
Keynote	Frederic Bushman	Integration of retroviral DNA: Mechanism and consequences *
2	Robert Craigie	Nucleoprotein intermediates in HIV-1 DNA integration *
	Vincent Parissi	Functional architecture of the HIV-1 integration complex integrase-DNA…“The good couple”
	Duane Grandgenett	HIV-1 synaptic complexes: molecular mechanisms associated with concerted integration
	Mamuka Kvaratskhelia	Dynamic modulation of HIV-1 integrase structure and function by cellular LEDGF protein as a therapeutic target *
	Stephen Hare	A reversible charge-charge interface involving the HIV-1 IN N-terminal domain is essential for high affinity interaction with LEDGF
3	Marc Ruff	Structural basis for HIV-1 DNA integration in the human genome (short talk)
	Akram Alian	A catalytically active complex of HIV-1 integrase bound to a viral DNA substrate that binds anti-integrase drugs
	Monica Roth	Moloney murine leukemia virus integrase (M-MuLV IN): Structural studies of the N-terminal domain and effects of HDAC inhibitors on C-terminal domain mutants *
	Mark Andrake	SAXS solution structure of a full length retroviral integrase
4	John Coffin	Sequence preferences for retroviral DNA integration
	Eric Poeschla	LEDGF/p75 proteins with alternative chromatin tethers are functional HIV-1 cofactors
	Dan Voytas	Targeting integration to heterochromatin by the yeast Ty5 retrotransposon
5	Richard Benarous	Identification of cellular co-factors of HIV-1 integrase by protein-protein interactions revealed by two-hybrid screens
	Mark Muesing	Towards revealing the HIV-1—host interactome
	Stephen Goff	Host proteins affecting Moloney murine leukemia virus: Interactions with the preintegration complex or integrase
	Ganjam Kalpana	Multiple roles of INI1 during HIV-1 replication
	Youichi Suzuki	Functional disruption of the MoMLV PIC by a cellular kinase, VRK
6	Michael Miller	HIV-1 integrase inhibitors: - From the bench to the clinic, and back again *
	Jean-François Mouscadet	In silico study suggests that raltegravir-resistant mutations modify the DNA recognition properties of HIV-1 integrase
	Jonathan Leis	Defining the DNA substrate binding sites of HIV-1 integrase
	Ira Dicker	The terminal (catalytic) adenosine of the HIV LTR controls the kinetics of binding and dissociation of HIV integrase strand transfer inhibitors (short talk) *
7	W. Edward Robinson	Resistance to HIV-1 integrase inhibitors reveals several phenotypes
	Nouri Neamati	A new paradigm for integrase inhibition: Blocking enzyme function without directly targeting the active site
	Yves Pommier	Drugging the active site of HIV-1 integrase *
8	Anna Cereseto	Development toward intranuclear visualization of HIV-1 pre-integration complexes
	Alan Engelman	Characterization of PWWP domain residues critical for LEDGF/p75 chromatin-binding and HIV-1 infectivity
	Zeger Debyser	Transportin-SR2 imports HIV into the nucleus
	Samson Chow	Functional role of interaction between HIV-1 integrase and Nucleoporin 153 in nuclear import of HIV-1
	Takao Masuda	Functional evaluation of the interaction between HIV-1 integrase and its interactor GEMIN2
9	Henry Levin	Retrotransposon Tf1 is targeted to Pol II promoters by transcription activators *
	Stephen Hughes	Integration of ASLV linear viral DNAs with aberrant ends
	David Garfinkel	Ty1 integrase and insertional mutagenesis in Saccharomyces cerevisiae
10	Kushol Gupta	Small-angle scattering studies of HIV-1 integrase-LEDGF-DNA complexes (short talk)
	Michael Katzman	Development of a high-throughput assay to identify novel stimulators of the nonspecific endonuclease activity of HIV-1 integrase and proof-of-concept retrovirus inhibition by known integrase stimulator (IS) compounds
	William Reznikoff	Probing diketoacid contacts using the Tn5 transposition system
	Stéphane Emiliani	A large-scale yeast two-hybrid screening approach to identify new host co-factors of HIV-1 integrase

^1^ Talks highlighted in main text denoted by *.

#### 3.3.1. Session 1: Integration of Retroviral DNA: Mechanism and Consequences

Given the 7 year gap since the Paris meeting, several field advancements had since transpired. Key among these was Frederic Bushman’s high-resolution mapping of sites of HIV-1 integration in the human genome [73]. In this seminal 2002 study, asymmetric DNA linkers were ligated with genomic DNA isolated from infected cells that had been sheared via restriction endonuclease digestion. Following nested PCR using LTR- and linker-specific primers, DNA fragments were cloned into plasmid DNA and, following bacterial transformation, the resulting plasmid minipreps were sequenced using dideoxy sequencing to identify LTR-host junctions. With the publication of the draft human genome in 2001, sites of HIV-1 integration could be mapped with nucleotide-level precision to genomic features such as genes, promoter regions, gene-dense regions, transcriptional activity, etc. The results of the 2002 paper first showed that HIV-1 favors active genes and gene-dense regions of chromatin for integration. In his keynote talk, Dr. Bushman described the use of pyrophosphate-based next-generation sequencing, which significantly increased the number of mapped integration sites from several hundred per experiment to tens of thousands. Several conclusions could be drawn from these richer integration-site datasets, including that integration tracked with histone post-translational modifications associated with active genes and that integration favored sites on outward-facing DNA major grooves on nucleosomes [75]. The Bushman laboratory was also the first to demonstrate a role for LEDGF/p75 in targeting HIV-1 PICs to active genes for integration [106].

#### 3.3.2. Session 2: HIV-1 Nucleoprotein Complexes and Interactions with LEDGF

As mentioned earlier, detailed characterization of PICs isolated from retrovirus-infected cells is exceedingly challenging. Robert Craigie presented a novel approach to constructing IN-LTR complexes capable of concerted integration activity that were also sufficiently stable for biophysical analyses. This approach relied on linear DNA substrates of sufficient length (~500–1500 bp) that contained U5 and U3 end sequences. Such DNAs successfully assembled with IN into a stable synaptic complex (SSC) that resisted challenge with 0.5 M NaCl and could be isolated from agarose gels following electrophoresis [9,107]. DNase I-based footprinting indicated that an IN tetramer protected ~20 bp of each DNA end. Integration into target DNA occurred sequentially; once the first end had integrated, the second end became joined at greater than 95% efficiency, avoiding the unwanted consequence of half-site integration. Although the recombinant SSCs supported concerted integration activity at a level rivaling PICs isolated from virus-infected cells, one issue was scalability for structural biology studies. In ensuing years, the Craigie lab described heterologous solubility-enhancing protein domains that, when fused to IN [91,108], yielded HIV-1 intasomes with short oligonucleotide substrates that were amenable to structural analysis via single-particle cryogenic electron microscopy (cryo-EM) [89,90,91].

Mamuka Kvaratskhelia characterized HIV-1 IN as a highly dynamic protein in equilibrium between lower-order monomer/dimer forms and a higher-order tetramer form. Moreover, LEDGF/p75 binding stabilized the tetrameric form of HIV-1 IN [109]. Mass spectrometric-based protein footprinting identified IN tetramer interfaces important for catalytic activities and high-affinity LEDGF/p75 binding. Dr. Kvaratskhelia suggested that the highly dynamic behavior of IN subunit-subunit interactions could be exploited as a novel antiviral target and presented preliminary studies with an acetylated molecule that engaged IN at the CCD-CCD dimer interface and restricted protein subunit exchange [110]. These seminal findings would set the stage for the subsequent discovery of allosteric IN inhibitors (ALLINIs) that engage the CCD-CCD dimer interface at the site for LEDGF/p75 binding, the consequences of which impart catastrophic, aberrant IN hyper-multimerization (reviewed in ref. [11]).

#### 3.3.3. Session 3: Structural Studies of Retroviral INs

Mo-MLV/γ-retroviral intasomes have to date resisted detailed structural analyses. Thus, more so than for other well-studied retroviral species, the structures of isolated Mo-MLV IN fragments/domains were highly topical. Monica Roth described the 3-dimensional structure of Mo-MLV IN residues 1-106 that included the NED in addition to the NTD (Figure 3) [111]. These studies revealed a protein dimer with, as expected, Zn^2+^ coordination via the conserved residues of the HHCC sequence motif. Several site-directed mutants of the IN CTD were shown to display delayed viral passage but maintain WT levels of IN activities in in vitro integration assays. The delay in viral passage, moreover, could be overcome by the addition of histone deacetylase inhibitors to the cell cultures, which was determined to enhance expression from unintegrated viral DNA [112]. More recent studies have clarified that retroviral DNA is rapidly heterochromatinized following nuclear entry, a process that significantly restricts viral gene expression prior to integration [113,114,115].

#### 3.3.4. Sessions 6 and 7: HIV-1 IN Inhibitors and Drug-Resistance

The hunt for the first clinically effective HIV-1 IN inhibitor concluded with FDA approval of raltegravir in 2007, which was sold by Merck under the tradename Isentress (Table 1). Michael Miller presented an overview of this Herculean accomplishment, including a discussion of Merck’s initial DKA compounds [18] and ensuing drugs such as the L-870,812 napthyridine carboxamide, which inhibited simian-HIV (SHIV) replication in the macaque model [116]. Dr. Miller also described some of the initial genetic pathways identified from clinical trials wherein amino acid changes in IN conferred substantial resistance to RAL [117]. The novel INSTI MK-2048 was, moreover, described as having a higher barrier to the development of resistance compared to RAL, which is a key trait that distinguishes the second-generation INSTIs from RAL and EVG. 

Ira Dicker presented an important contribution related to the terminal deoxyadenylate of the invariant LTR CA motif in INSTI binding and inhibition. Pioneering work from Merck previously revealed that predecessor DKA compounds effectively bound IN-LTR complexes as compared to IN alone [105], but details of this key INSTI-IN-LTR interaction were lacking due to the complete absence of IN-LTR structures at the time. Dr. Dicker employed a scintillation proximity assay with tritiated INSTI BMS-641493 to investigate the role of LTR base residues in INSTI binding, concluding that the invariant adenosine slowed the rate of INSTI association and also the rate of INSTI dissociation from the nucleoprotein complex [118]. Later, studies of PFV intasome structures revealed that INSTIs disarm the nucleoprotein complex by displacing the adenosine from its committed position, which removes the 3′-oxygen atom required for strand transfer activity from the IN active site [21]. The INSTIs, moreover, form novel contacts with the supplanted adenosine [119]. The solution-based measurements presented by Dicker and colleagues are completely consistent with the structure-based results from PFV intasomes that would be elucidated in advance of the next IN conference. 

Yves Pommier analyzed HIV-1 IN mutant data from patients who experienced therapeutic failure during treatment with RAL or EVG [120]. Seven mutant variants were expressed and purified as recombinant IN proteins and compared to WT HIV-1 IN. All of the mutant IN proteins were partially impaired for strand transfer activity, with the Q148K mutant also significantly impaired for 3′ processing activity. Both compounds exhibited comparable resistance profiles; of the tested mutants, Q148K and T66I conferred the highest levels of resistance, while S153Y conferred comparatively greater resistance to EVG as compared to RAL. Importantly, studies such as this one demonstrated comparable cross-resistance of IN mutations to both RAL and EVG, highlighting the need to develop next-generation compounds that would hopefully address the issue of common cross-resistance for virological failure to INSTIs.

#### 3.3.5. Session 9: Retrotransposon INs

The LTR-retrotransposon Tf1 displays many properties associated with retroviruses, thus providing a highly tractable genetic system for exploration. Similar to the known integration-targeting biases of HIV-1 and Mo-MLV, which highly prefer active genes and promoter regions, respectively [73,121], Tf1 integration occurred upstream of open reading frames. Henry Levin presented evidence that Tf1 integration was targeted specifically to the promoter regions of Pol II-transcribed genes [122,123]. Using a plasmid-based integration-targeting system in cells, Tf1 IN was shown to interact with transcription factor Atf1p to direct integration as a tethering factor to the *fbp1* gene promoter. Given the immense timeframe of cell–retrotransposon coevolution, studies of Tf1 and other species of retrotransposons have greatly facilitated our appreciation of how retroelements leverage interactions with host cell factors to direct integration into mutually beneficial regions of chromatin [124,125]. 

### 3.4. The Fourth International Conference

The fourth conference was held 4–7 October 2011 in Siena, Italy, at the University of Siena. The title was “4^th^ International Meeting on Retroviral Integration”. The main organizers were Zeger Debyser, Maurizio Botta, and Frauke Christ; Frederic Bushman, Alan Engelman, and Yves Pommier as US delegates filled out the organizing committee. There were 126 registrants.

This conference expanded on the theme to select short talks from submitted abstracts, and thus there were overall a large number of talks at this conference (Table 5). Each session additionally had a plenary talk presented by the session chair, as well as talks from invited speakers. The final session of the meeting, called “Mini-symposium on gene therapy” (session 9), consisted of 11 talks and was in essence a meeting within a meeting. The keynote talk on the first evening of the conference was given by Michael Miller, who presented an overview of Merck’s INSTI program, an update on the contributions of IN mutations to clinical drug resistance, and discussed possibilities of using INSTIs as part of pre-exposure prophylaxis regimens to prevent HIV-1 acquisition. 

**Table 5 viruses-16-00604-t005:** Talks presented at the 2011 Siena meeting ^1^.

Session	Speaker	Title
Keynote	Michael Miller	*INSTI development program at Merck and IN drug resistance changes* ^2^
1	Peter Cherepanov	The mechanism of HIV integration and its inhibition by strand transfer inhibitors: lessons from x-ray crystallography using a convenient model system (plenary talk) *
	Kushol Gupta	Small-angle scattering studies of retroviral integrase-DNA complexes
	Mark Andrake	Architecture of a full-length retroviral integrase monomer and dimer, revealed by small angle x-ray scattering and chemical cross-linking (short talk)
	David Langley	HIV-1 integrase: structure and function (short talk)
	Sherwin Montano	Crystal structure of the bacteriophage Mu transpososomes
	Barbara Capecchi	Antibody-mediated protection against HIV infection using Env vaccines
2	Alan Engelman	Integrase biochemistry and HIV-1 replication (plenary talk)
	Robert Craigie	*Nucleoprotein intermediates in HIV-1 integration; hyperactive HIV-1 IN proteins*
	Mamuka Kvaratskhelia	Modulation of HIV-1 integrase structure and function by LEDGF/p75
	Vincent Parissi	Role of the HRAD51 DNA repair protein in the HIV-1 integration and post integration repair steps (short talk) *
	Karine Pradeau	Post-translational modification and functional analysis of the HIV-1 IN/LEDGF complex produced in mammalian cells (short talk)
	Awatef Allouch	KAP1 targets acetylated integrase and inhibits HIV-1 integration (short talk)
3	Zeger Debyser	Cofactors of HIV integration from target validation to drug discovery (plenary talk)
	Eric Poeschla	LEDGF dominant interference: what is it telling us about the post-entry journey of HIV-1? *
	Ganjam Kalpana	INI1/HSNF5-interaction defective HIV-1 IN mutants exhibit impaired reverse transcription and integration in vivo (short talk)
	Marc Lavigne	Identification of new partners of the LEDGF/p75 protein, an essential cofactor of HIV-1 integrase
	Nicolas Soler	Characterization of HIV-1 integrase complex formation during T-cell infection reveals dynamic association with cellular cofactors (short talk)
4	Anna Cereseto	Retro-images from infected cells (plenary talk) *
	Thomas Hope	Exploring the relationship between HIV reverse transcription, trafficking, uncoating, and nuclear import
	Gloria Arriagada	MuLV capsid, SUMoylation, and TRIM5alpha recognition
	Stephanie De Houwer	Transportin-SR2 and HIV-1 integrase, partners for HIV nuclear import (short talk)
	Ross Larue	Biochemical analysis of HIV-1 integrase interactions with transportin-SR2 reveals functional protein-protein contacts (short talk)
	Christine Di Primio	Visualization of HIV-1 DNA in infected cells using a new fluorescent virus-based reporter system (short talk)
	Alberto De Iaco	TNPO3 promotes HIV-1 infection at a step after nuclear entry (short talk)
	Samson Chow	Role of HIV-1 integrase during uncoating and reverse transcription
5	Yves Pommier	Overcoming raltegravir resistance (plenary talk)
	Mark Underwood	Effects of accumulating RAL signature and secondary mutations on dolutegravir (DTG, S/GSK1349572) activity *
	Romas Geleziunas	Next generation HIV-1 integrase strand transfer inhibitors
	Ira Dicker	Probing the role of Mg in integrase strand transfer catalysis and in the binding of first and 2nd generation STIs
	Chris Pickford	Pre-clinical evaluation of HIV replication inhibitors that target the integrase-LEDGF/p75 interaction
	Louie Lamorte	Discovery of a novel non-catalytic site integrase inhibitor
	Maurizio Botta	Studies on the inhibition of HIV-1 integrase
	Nouri Neamati	Design of cell permeable nanoneedles as HIV-1 integrase inhibitors
	Francesca Morreale	Computational approaches for the identification of small molecules as inhibitors of HIV-1 IN-LEDGF/p75 interactions (short talk)
	Christophe Marchand	Novel HIV-1 integrase inhibitors targeting the interface of the N- and C-terminal domains and overcoming resistance to strand transfer inhibitors (short talk)
6	Mark Wainberg	Dolutegravir selects for a R263K mutation in HIV-1 subtype B and AG but not in subtype C viruses (plenary talk)
	Francesca Ceccherini-Silberstein	New insights of HIV resistance to integrase inhibitors
	Paradise Madlala	The influence of genetic variation of transportin-SR2 (TNPO3) gene on susceptibility to HIV-1 infection and disease outcomes (short talk)
	Marie-Line Andreola	The addition of the integrase mutation T97A to the primary mutations Y143R/C strongly increases the in vitro resistance to RAL and rescues the catalytic defect conferred by Y143R (short talk)
7	Suzanne Sandmeyer	Position specificity of a retrotransposon integrase (plenary talk) *
	Henry Levin	High-throughput sequencing of retrotransposon integration provides a genome-wide profile of target activity
	Zoltan Ivics	Genetic engineering with sleeping beauty transposons
	Andrea Cara	Development and use of integrase defective lentiviral vectors for immunization
	Monica Roth	Rescuing MLV p12 mutants with DNA tethering domains (short talk)
	Duane Grandgenett	Historical aspects of retrovirus integrase research
9	Frederic Bushman	Transformation and clonal expansion during human gene correction using retroviral vectors (plenary talk)
	Luigi Naldini	*Targeted integration for gene therapy vectors* (plenary talk)
	Fulvio Mavillo	Defining the lentiviral integrome IN (plenary talk) *
	Christof Von Kalle	Insertional repertoires of targeted and non-targeted vectors
	Mauro Giacca	HIV-1 integrase stability and nuclear topography regulate viral DNA integration in primary CD4+ T cells
	Sylvia Kaulfuss	Advantages of an expression-optimized prototype foamy virus pol for vector system development (short talk)
	Corinne Ronfort	Gene expression profiling and cell signaling pathways modified by retroviral integration (short talk)
	Pascale Lesage	Implication of the AC40 subunit of RNA polymerase III in Ty1 integration (short talk)
	Valentina Poletti	Genome-wide definition of transcriptionally active regulatory elements in human stem cells by retroviral scanning (short talk)
	Paul Lesbats	Functional coupling between HIV-1 integrase and the SWI/SNF chromatin remodeling complex for efficient in vitro integration into stable nucleosomes (short talk)
	Rik Gijsbers	The use of LEDGF/p75 chimera to retarget lentiviral integration
	Stéphane Emiliani	A large-scale yeast two-hybrid screening approach to identify new host co-factors of HIV-1 integrase

^1^ Talks highlighted in main text denoted by *. ^2^ Lacking specific titles, italics denote subject(s) covered.

#### 3.4.1. Session 1: Structural Biology

2010 had been a banner year for IN structural biology efforts. The Cherepanov lab in 2009 published that PFV IN was highly soluble and, moreover, displayed efficient concerted integration activity using comparatively short LTR donor DNAs with little evidence for half-site integration activity [13]. These results set the stage for the ensuing X-ray crystal structures. Peter Cherepanov presented the structure of the PFV intasome composed of IN bound to 19-mer pre-cleaved LTR substrate, which diffracted X-rays to 2.9 Å resolution [21]. The asymmetric unit contained an IN dimer complexed with a viral DNA molecule, and a pair of symmetry-related dimers formed a tetrameric structure whose overall architecture was described as a dimer of dimers. The outer dimers of the tetramer were built around the canonical CCD dimer and, as predicted from the initial HIV-1 IN CCD structure, only one of these two active sites was intimately involved with the viral DNA end and was thus relevant to the DNA-cutting and -joining steps of integration. The inner dimer, closely intertwined with both LTR ends, was novel. The NTDs of the two inner dimer molecules interacted with the CCDs of the opposing IN subunit. This “domain-swapped” orientation was completely consistent with early biochemical results that indicated the NTD of one IN molecule acted in trans with the CCD of a second IN molecule within the functional HIV-1 IN multimer [60]. Dr. Cherepanov also presented structures of INSTIs bound at the PFV IN active sites, which, as alluded to above, elucidated how the drugs disarm the nucleoprotein complex. Two common drug entities, co-planar heteroatoms and halo-benzyl side-chains, mediated key intasome interactions. As had been concluded from prior solution-based measures of DKAs with HIV-1 IN [126], drug heteroatoms engaged the DDE-coordinated divalent metal ions. The halo-benzyl groups intercalated with the penultimate G-C base pair of the LTR end to displace the 3′-deoxyadenylate residue from its committed position within the active site [21,119] (Figure 4, updated for HIV-1 structures with BIC). Crystal structures of WT and PFV IN mutants in the presence of INSTIs also helped to explain why specific drug-resistance IN changes arise in PLWH in clinical settings. However, given the limited extent of amino acid identity between the PFV and HIV-1 INs, future structural studies of HIV-1 intasomes, as well as of intasomes from the more closely related simian immunodeficiency virus from red-capped mangabeys, were necessary to fine-tune INSTI-IN active-site interactions, especially as they pertain to second-generation INSTI compounds (Figure 4) [90,127,128]. Finally, Cherepanov presented X-ray crystal structures of PFV intasomes bound to target DNA, as well as following covalent insertion of the viral DNA ends into the target DNA [129]. These structures elucidated significant expansion and compression of the target DNA major and minor grooves, respectively, which are required to accommodate the two scissile phosphodiester bonds within target DNA at the two IN active sites. The importance of this series of PFV intasome structures to the field of retroviral integration cannot be overstated. Although they were in the long run supplanted by superior primate lentiviral IN models, the PFV structures nevertheless clarified the INSTI mechanism of action and set the tone for new standards in IN structural biology. Structures of PFV intasome-mediated 3′ processing and strand transfer reactions as a function of time post-initiation via metal ion soaking in crystallo soon after provided descriptions of IN’s reaction mechanisms in unparalleled detail [65]. The tetrameric PFV IN architecture, with NTDs swapped between two IN molecules intimately engaged with the LTR ends, has, moreover, been seen across all subsequent retroviral intasome structures and is accordingly referred to as the “conserved intasome core” (CIC; reviewed in ref. [11]).

#### 3.4.2. Session 2: Biochemistry of Integration

The product of DNA strand transfer is a recombination intermediate with single-stranded DNA gaps flanking unjoined 3′-ends of host chromosomal target DNA (Figure 1). Numerous cellular factors have been implicated in the ensuing DNA-repair process, including flap endonuclease, DNA polymerase, and DNA ligase [130]. Vincent Parissi had previously shown that hRAD51, which plays a major role in homologous recombination, interacted with HIV-1 IN and inhibited its activity [131]. Dr. Parissi, in his talk, demonstrated that the formation of an active hRAD51 nucleofilament was required for optimal inhibition and that this process involved dissociation of HIV-1 IN-DNA complexes. In addition, stimulation of hRAD51 activity increased the endogenous DNA-repair process and inhibited cell-based HIV-1 integration [132]. Additional cellular proteins since implicated in the repair of the HIV-1 integration intermediate include Ku70 [133], Ataxia-telangiectasia mutated (ATM) kinase, DNA-dependent protein kinase (DNA-PK) [134], and Fanconi anemia factors FANCI and FANCD2 [135]. Understanding the molecular details of DNA repair of the HIV-1 integration intermediate may provide new approaches to antiretroviral therapy [136]. 

#### 3.4.3. Session 3: Cellular Cofactors of Retroviral Replication

Previous work had established that LEDGF/p75 is a chromatin-associated transcriptional co-activator [137] that binds IN and directs HIV-1 integration into active genes [102,106,138]. Early steps in HIV-1 replication, such as reverse transcription, occur within the confines of the viral core, and exposure of inner-core components to the cellular milieu is a subject of ongoing debate as researchers in the field pursue the molecular details of capsid uncoating and capsid remodeling that may accompany certain virus-ingress steps, such as nuclear import [139]. In his talk, Eric Poeschla investigated IN-LEDGF/p75 interactions during HIV-1 infection through expression of novel LEDGF/p75 IN-binding domain (IBD) fusions with green fluorescent protein (GFP) in susceptible target cells [140]. Because the cytoplasmically located LEDGF/p75 fusion proteins restricted HIV-1 infection, these authors concluded that core-associated HIV-1 IN must be exposed to the action of the fusion proteins in the cell cytoplasm. Moreover, combining fusion protein expression with LEDGF/p75 depletion via RNA interference virtually eliminated HIV-1 infection. These latter results further highlighted the search for inhibitors of the interaction between LEDGF/p75 and HIV-1 IN [141].

#### 3.4.4. Session 4: Trafficking and Nuclear Import

Detailed analyses of HIV-1 cytoplasmic trafficking, nuclear import, and intranuclear trafficking to sites of viral DNA integration continue to be actively pursued in the fields of HIV-1 molecular and cellular biology. Key to such measurements in virus-infected cells is microscopy-based tracking of fluorescently labelled viral particles, work that was pioneered by Thomas Hope and colleagues [142]. In her plenary talk, Anna Cereseto described the development of fluorescently labelled HIV-1 to detect the migration of single viral particles into the cell nucleus [143,144]. Using this system, Dr. Cereseto concluded that HIV-1 PICs access cell nuclei by an active transport mechanism and that nuclear actin may facilitate post-nuclear PIC trafficking. These early studies helped to drive fluorescence-based measures of HIV-1 trafficking and nuclear import that are now much more commonplace among laboratories [145]. 

#### 3.4.5. Session 5: Drug Discovery

As discussed above, clinical resistance to RAL and EVG was not uncommon, and changes elicited in response to one inhibitor generally caused cross-resistance to the other [146]. GlaxoSmithKline pioneered the development of the second-generation inhibitor DTG, and Mark Underwood reported results of DTG inhibition in in vitro integration assays, as well as drug-susceptibility for clinical samples derived from 18 adults who had demonstrated incomplete viral suppression in response to RAL-based regimens [147,148]. Three pathways to RAL resistance, each involving a change of IN amino acid residue Tyr143, Gln148, or Asn155, were known [117]. DTG retained nearly full activity against clinical isolates whose main resistance changes involved Tyr143 or Asn155. Isolates containing IN changes G140S/Q148H and G140S/Q148R, by contrast, conferred approximately 4- and 13-fold resistance, respectively, to DTG. Studies such as this one defined the types of RAL resistance-conferring changes that additionally conferred resistance to second-generation INSTIs, which critically informs the ongoing rollout of regimens containing second-generation INSTIs to treat PLWH [19].

#### 3.4.6. Session 7: Other Retroviruses and Retrotransposons

Ty3 displays exquisite target-site specificity, integrating its genome within one or two nucleotides of the transcription-initiation sites of genes that are transcribed by RNA polymerase III [70]. In her plenary talk, Suzanne Sandmeyer described the development of an in vitro biochemical system to reconstitute the specificity of Ty3-targeted integration. The system leveraged a synthetic fusion protein of transcription factor (TF) IIIB subunits Brf1 and TBP. This in vitro system delineated TFIIIB domains targeted during Ty3 retrotransposition and also highlighted the central role of Ty3 IN in this process [149].

#### 3.4.7. Session 8: Mini-Symposium on Gene Therapy

Lentiviral vectors based on HIV-1 were being developed to treat human genetic disorders, and Fulvio Mavillo, in his plenary talk, described experiments to investigate stable gene transfer and integration site monitoring of allogeneic T cells after donor lymphocyte infusion [150]. Comparison of integration events to matched controls using CD34+ hematopoietic stem progenitor cells demonstrated that integration clustered within chromatin regions of active promoters and regulatory elements in cell-type-specific manners. Post-fusion analyses revealed no evidence for integration-related clonal expansion, but loss of cells when integration events interfered with RNA post-transcriptional processing. Lentiviral vectors continue to be convenient and efficient tools through which to transfer genes into human cells [151], and measurements of vector-related cellular expansion continue to be a critical part of the evaluative process used to monitor potential adverse side effects from integration-competent viral vectors. 

### 3.5. The Fifth International Conference

The fifth conference in the series, which convened 23–26 October 2014, was held in Pacific Grove, California, USA, at the Asilomar conference grounds. The title was “5^th^ International Conference on Retroviral Integration” (Table 6). The organizers were Samson Chow, Sherly Mosessian, Nouri Nemati, and Shaojun Zhu. There were 58 registrants.

**Table 6 viruses-16-00604-t006:** Talks presented at the 2014 Asilomar conference ^1^.

Session	Speaker	Title
Keynote	Alan Engelman	Integrase host cofactors: Unanticipated antiretroviral bedfellows *
1	Robert Craigie	Retroviral integrase: Activities and structure
	Michael Miller	Anti-IN inhibitors: Clinical experience and new drug development
2	Akram Alian	Another piece in the integrase multimerization puzzle: The first monomeric integrase core domain structure
	Duane Grandgenett	Structural biology of kinetically stabilized RSV and HIV-1 synaptic complexes produced with integrase strand transfer inhibitors
	Min Li	Outer integrase subunits in the intasome are dispensable for catalysis of integration and a “magic” peptide that enhances HIV-1 integrase *
	Mark Andrake	Multimerization properties of retroviral integrases
	Marc Ruff	The HIV-1 pre-integration complexes: Structure, function and dynamics
	Mamuka Kvaratskhelia	Structure and function of retroviral integrases as a therapeutic target
	Kellie Jurado	Characterization of HIV-1 particle maturation defect caused by allosteric integrase inhibitors (selected from abstracts)
	Samson Chow	Characterizing the interaction between HIV-1 IN and CA assemblies
3	Carlos Casiano	Beyond HIV-1 integration: Emerging roles of LEDGF/p75 in cancer and autoimmunity
	Anna Cereseto	3D analysis of retrovirus-nucleus interactions
	Ganjam Kalpana	An essential role of integrase binding protein INI1/hSNF5 in HIV-1 post-transcriptional mechanisms leading to assembly
	Eric Poeschla	TALEN knockout of the HIV-1 integration cofactor LEDGF/p75
	Anais Jaspart	Phosphorylation of HIV-1 integrase by GCN2 (selected from abstracts) *
4	Mark Underwood	HIV-1 primary and secondary integrase mutations: Dolutegravir clinical response, and effects on DTG, raltegravir (RAL), and elvitegravir (EVG) resistance and replication capacity *
	Yves Pommier	Novel INSTIs to overcome drug resistance mechanisms
	Zeger Debyser	Novel pleiotropic roles of HIV integrase revealed by LEDGINs and integrase polymorphisms
	Richard Benarous	Resistance analysis with HIV-1 integrase-LEDGF allosteric inhibitors that effect virion maturation but do not influence packaging of a functional RNA genome (selected from abstracts)
	Philippe Cotelle	2-Hydroxyisoquinoline—1,3(2H, 4H)—diones (HIDs), novel inhibitors of HIV integrase with a high barrier to resistance
	Ira Dicker	A simple and accurate in vitro method for predicting serum protein binding of HIV integrase strand transfer inhibitors
	Nouri Neamati	Discovery of first-in-class inhibitors of HIV-1 integrase-HSP90 interaction
5	Marc Lavigne	Role of DNA and chromatin structure in HIV-1 integration
	Vincent Parissi	Regulation of retroviral integration by chromatin and intasome structures
	Stephen Hughes	Specific HIV integration sites are linked to the clonal expansion and persistence of infected cells in patients *
	Henry Levin	Analysis of 1-million independent HIV-1 integration sites identifies a link with mRNA splicing
	Monica J. Roth	MLV integration site selection
	Frederic Bushman	Retroviral DNA integration in human gene therapy
6	Suzanne Sandmeyer	Parsing the determinants of extreme integration specificity
	Karen Beemon	Targets of integration of ALV-J in chicken hemangiomas *
	Donald Kohn	Clinical applications of integrating vectors for gene therapy
	Emmanuelle Six	Tracking the dynamic of hematopoietic progenitors through integration site analysis in gene therapy trials
	Richard Gabriel	Integration of retroviral vectors in gene therapy—understanding and avoiding severe side effects
	Szilvia Solyom	Massive somatic L1 retrotransposition occurs early during gastrointestinal tumorigenesis

^1^ Talks highlighted in main text denoted by *.

#### 3.5.1. “IN Host Cofactors: Unanticipated Antiretroviral Bedfellows” Keynote Presentation

As touched upon in the Introduction (Section 1), mutations in HIV-1 IN disrupt virus replication in different ways. Class I IN mutations specifically inhibit integration, while class II mutations impede proper virion morphogenesis. The conical HIV-1 core is composed of an outer shell of CA that encases the viral ribonucleoprotein complex (vRNP), which is composed of viral RNA and viral proteins nucleocapsid, RT, and IN. Upon fixation and staining of microtome-thin sections with heavy metals, vRNPs appear comparatively electron-dense by transmission electron microscopy. With class II IN mutant virions, the electron density appears outside the CA shell, most usually in association with the viral membrane, the consequences of which significantly reduce DNA synthesis levels after virus infection [41,152]. Alan Engelman, in his keynote talk, highlighted the finding that ALLINIs result in the generation of the same type of morphologically defective particles that are associated with class II IN mutations. Also called NCINIs (for non-catalytic site IN inhibitors) or LEDGINs (for LEDGF-IN inhibitors), ALLINIs were discovered via a high-throughput screening for inhibitors of IN 3′ processing activity [153] and were shown to inhibit the IN-LEDGF/p75 interaction [141]. Although inhibition of IN-LEDGF/p75 binding was initially thought to underlie the antiviral mechanism, Engelman’s work clarified that exposing HIV-1 to the drugs during virus production inhibited subsequent viral infection more potently than did treating the infected cells themselves during the early phase of HIV-1 infection when integration occurs. Moreover, because ALLINI potency during the early infection phase was increased (and not decreased) by depleting the cellular content of LEDGF/p75, it seemed unlikely that inhibition of IN-LEDGF/p75 binding would contribute much at all to overall antiviral potency [152]. Additional research has clarified that ALLINI binding to the HIV-1 IN CCD dimeric interface provides a template for a secondary binding site for the CTD of a separate IN multimer [154,155]. In this way, ALLINIs serve as molecular glue to cascade the formation of linear and branch-chain polymers of HIV-1 IN [156], the consequences of which inhibit IN-RNA interactions and yield morphologically defective virus particles [157]. As of this writing, the ALLINI STP0404, a.k.a. pirmitegravir, has advanced to phase II clinical trials [158].

#### 3.5.2. Session 2: Basic Biology and Structure

Purified recombinant HIV-1 IN displays sparingly limited solubility at the near-isotonic salt concentrations favored for structural biology studies. To circumvent this shortcoming, Min Li and Robert Craigie previously appended a small non-specific DNA-binding protein from *Sulfolobus solfataricus*, called Sso7d, to the N-terminus of HIV-1 IN, which significantly improved both HIV-1 IN solubility and concerted integration activity [108] (Table 5). In his talk, Dr. Li described that a 20-mer P5 peptide derived from the DNA-binding AT-hook region of LEDGF/p75 yielded results highly similar to those found with the Sso7d fusion partner. The majority of HIV-1 intasome structures, including comparatively high-resolution structures with INSTIs bound, have since leveraged the Sso7d-IN fusion protein [89,90,128] (Figure 4). Demonstrating the versatility of the enhanced solubility approach, the cryo-EM structure of P5-IN bound to 32-mer pre-cleaved LTR DNA was subsequently resolved to 4.7 Å resolution [91].

#### 3.5.3. Session 3: Cellular Cofactors and IN’s Pleiotropic Actions

Anais Jaspert reported that infection by HIV-1 initiates an acute decrease in cellular translation. Infection was determined to lead to phosphorylation of GCN2, a cellular Ser/Thr kinase that was shown to interact with HIV-1 IN [159,160]. Kinase reactions conducted in vitro revealed that IN residue Ser255 was a major site of phosphorylation and that the IN proteins from other retroviral species were also phosphorylated. HIV-1 IN S255 mutant viruses displayed increased infection rates that correlated with an increase in viral DNA integration. Infectivity of Mo-MLV was also higher in cells knocked-out for GCN2, suggesting a conserved mechanism to control retrovirus replication. 

#### 3.5.4. Session 4: IN Inhibitors: Clinical Efficacy, Resistance, and New Inhibitor Development

Mark Underwood described the use of tritiated INSTI compounds in IN-LTR binding assays and the use of molecular modeling to understand the molecular basis of the favored drug-resistance profile of DTG as compared to the predecessor RAL and EVG compounds [161,162]. Kinetic studies of inhibitor dissociation from WT and mutant IN/DNA complexes showed that DTG had a significantly lower dissociation rate [dissociative half-life (t_1/2_) = 71 h] compared to RAL or EVG (t_1/2_ = 8.8 and 2.7 h, respectively). The longer residency time of DTG at the IN active site indicated that dissociative half-life is a major factor contributing to the efficacy of second-generation INSTIs. Consistent with this notion, a more recent study determined dissociative half-lives of 163 h, 96 h, 10 h, and 3.3 h for BIC, DTG, RAL, and EVG, respectively [163]. Other recent research has indicated that a primary determinant of INSTI resistance occurs via detuning of the divalent metal ion cluster at the IN active site. Because the intasome has to perform only one set of 3′ processing reactions and one set of strand transfer reactions per infectious cycle, the virus can tolerate some modest loss of IN catalytic function [164]. Thus, changes that decrease the strength of the magnesium ion cluster at the active site may preferentially result in significantly lower INSTI dissociative half-lives while maintaining sufficient IN catalytic function for viral DNA integration [127,128]. 

#### 3.5.5. Session 5: Integration-Site Selection: Case and Consequence

HIV-1 integration underlies the formation of a latent reservoir of cells infected with replication-competent viruses that persists in PLWH despite prolonged periods of anti-retroviral therapy and, accordingly, prevents HIV cure [165]. Strategies for HIV cure in turn rely profoundly on the characterization of reservoir cells and how the associated population of HIV-1 proviruses may change over time during therapy. Stephen Hughes described a modification of the LM-PCR assay to map sites of integration. If infected cells had divided post HIV-integration, random shearing of genomic DNA using sonication could detect the same provirus on multiple DNA molecules via the counting of unique breakpoints at the sites of linker ligation. Charles Bangham had devised this technique to monitor human T-lymphotropic virus 1 (HTLV-1) infection, which produces large clones of virus-infected cells [166]. Using this technique, Hughes and colleagues observed that approximately 40% of HIV-1 integration events occurred in clonally expanded cells and that integrations in particular loci, such as intronic regions of *MKL2* and *BACH2*, significantly tracked with clonal expansion [167]. The results of this study and of a parallel study performed at the same time [168] indicated that integration into growth-promoting genes may underlie clonal expansion and thus significantly help to mold the latent reservoir. Subsequent work has indicated that immune pressure also exerts significant influences, leading to clonal expansion and retraction [169], and that integration into only a handful of human genes in patients on long-term therapy is over-represented compared to the distribution that would be expected by random chance [170,171]. Most recently, it has been proposed that proviruses over long periods of time may accumulate in gene-sparse regions (a.k.a. “gene deserts”), indicating potential pathways to perhaps alleviating the burden of long-term antiretroviral treatment based on patterns of resident, intact proviruses [172].

#### 3.5.6. Session 6: Retrovirus-Based Vectors and Retrotransposons 

Avian leukosis virus subgroup J (ALV-J) causes hemangiomas and myeloid tumors in chickens and integrates into the chicken genome with little preference for genomic features such as genes or promoters. In general, ALV causes cancers by insertional mutagenesis mechanisms whereby integration into or nearby particular genes results in a highly significant cell growth advantage and tumorigenesis. Karen Beemon investigated the pathogenesis of infection in birds by using high-throughput DNA sequencing to analyze proviral integration sites in tumors [173], an approach that uncovered expanded clones with integration sites in the *MET* proto-oncogene gene in two of five hemangiomas. ALV-J integrations within *MET*, moreover, induced strong overexpression of *MET* mRNA. MET is a receptor tyrosine kinase implicated in numerous human cancers [174].

### 3.6. The Sixth International Conference

This conference, entitled “6^th^ International Conference on Retroviral Integration”, was held 18–21 September 2017 in Bordeaux, France, at the Cite Mondiale Convention Center. The meeting was organized by Marie-Line Andreola, Mathieu Metifiot, and Vincent Parissi, with input from advisory board members Richard Benarous, Samson Chow, Zeger Debyser, Olivier Delelis, Alan Engelman, Patrice Gouet, Marc Lavigne, Pascal Lesage, and Marc Ruff. Anna Marie Skalka presented the keynote talk “The Integrase Moveable Feast” to start the meeting (Table 7). There were 72 registrants.

**Table 7 viruses-16-00604-t007:** Talks presented at the 2017 Bordeaux conference ^1^.

Session	Speaker	Title
Keynote	Anna Marie Skalka	The integrase moveable feast
1A	Min Li	HIV-1 assembles multiple stable synaptic complex (SSC) intasomes that are active for concerted integration in vitro *
	Akram Alian	The C-terminus of alpha helix-4 offers novel hotspots within the HIV-1 integrase core domain
	Alison Ballandras-Colas	Cryo-EM structures of the Maedi-visna virus intasome *
	Julien Batisse	The HIV-1 pre-integration complexes: Structures, functions and dynamics
1B	Duane Grandgenett	Assembly and functions of Rous sarcoma virus synaptic complexes containing integrase tetramers and octamers
	Daniela Lerner	Conserved but flexible: A new essential motif in the C-ter domain of integrase characteristic of group M
	Jacques Oberto	Flipping chromosomes in deep-sea Archaea
	Flore De Wit	Visualization of the human immunodeficiency virus type 1 cDNA by click chemistry
	Cyril Masante	Presentation of chemometec cell analysers
2A	Alan Engelman	Virus-host interactions that regulate HIV-1 integration *
	Vincent Parissi	Regulation of retroviral integration by Pol II transcription associated factors and chromatin structure
2B	Marc Lavigne	Regulation of HIV-1 integration and transcription by cellular DNA topology, consequences on viral replication and latency *
	Ganjam V. Kalpana	NMR structure of conserved Rpt1 domain of INI1/SMARCB1: A structural basis of HIV-1 IN-INI1 interactions and functional significance *
	Marie-Line Andreola	GCN2 acts as a restriction factor to multiple retroviral infections through phosphorylation of their integrase
2C	Karen Beemon	The FACT complex promotes avian leukosis virus integration
	Eric Mauro	Influence of histone tails on HIV-1 integration: structure-function and therapeutical approach
	Pascal Lesage	Molecular mechanisms and regulation of Ty1 retrotransposition integration site selection *
	Henry Levin	Transposable element integration rewires regulatory networks to protect cells from stress
3A	Zoltan Ivics	Transposase and host factor determinants of target cell selection by DNA transposons
	Wei Shao	New functions and updates of the Retrovirus Integration Database (RID) and its applications
	Goedele Maertens	Protein phosphatase 2a (PP2a) influences integration site selection of human T-lymphotropic virus type-1 (HTLV-1) *
3B	Dominique Van Looveren	Engineering next generation BET-independent MLV-vectors for safer gene therapy
	Paul Lesbats	Unraveling the link between foamy virus Gag nuclear trafficking and integration site selection
3C	Dalibor Miklik	Transcriptional start sites and enhancers are genomic loci permissive for long-term stable expression of proviruses *
	Gerlinde Vansant	LEDGINs retarget integration and hamper the establishment of a reactivation competent reservoir
	Stéphane Emiliani	Iws1 participates to the maintenance of HIV latency and is recruited during HIV transcription
Selected Talks	Thomas Gayraud	Understanding the role of primary resting CD4 T cells, nuclear organization and function, in HIV integration site selection and proviral transcription
	Francesca Di Nunzio	The nuclear pore complex orchestrates HIV-1 nuclear import and sculpts the chromatin landscape near integration sites
	Heng-Chang Chen	HIV expression, PEV and genome architecture
4A	Romina Quercia	ViiV Healthcare
	Zeger Debyser	Single virus imaging of LEDGIN-mediated inhibition of HIV replication
	Mark Andrake	A screen for inhibitors of HIV-1 integrase multimerization yields new allosteric candidates
	Mathieu Metifiot	Rational design of HIV integrase active site inhibitors with broad spectrum of action against resistant mutants
	Jolien Blokken	Inhibitors of the integrase-transportin-SR2 interaction block HIV nuclear import
	Mamuka Kvaratskhelia	Critical structural determinants for ALLINI-induced hyper-multimerization of HIV-1 integrase
4B	Ariberto Fassati	Digoxin reveals a functional connection between HIV-1 integration preference and T-cell activation
	Herve Fleury	Next-generation sequencing data for characterization of CTL epitopes in archived HIV-1 proviral DNA
	Samson Chow	Efficient identification of mutations conferring resistance to integrase inhibitors using a randomly mutated library of HIV-1 integrase
	Olivier Delelis	Novel mutations outside the integrase gene confer HIV-1 integrase strand-transfer inhibitors resistance *

^1^ Talks highlighted in main text denoted by *.

#### 3.6.1. Sessions 1A–B: Molecular, Structural, and Imaging Analyses of Retroviral Integration 

The development of solubility-enhancing HIV-1 fusion proteins Sso7d-IN and P5-IN paved the way for the creation of cryo-EM-based structures of HIV-1 intasomes. Working with Dmitry Lyumkis, Min Li described the cryo-EM structure of the HIV-1 strand transfer complex (STC) intasome constructed from Sso7d-IN and a hybrid LTR-target DNA substrate that modeled the product of IN strand transfer activity [89]. A tetrameric form of the STC intasome, which architecturally resembled the PFV intasomal IN tetramer, was resolved to ~3.5–4.5 Å, with the greatest resolution in and around the CIC. STC intasome assembly in the presence of the LEDGF/p75 IBD yielded a greater proportion of higher-order assemblies and a comparatively lower-resolution map for a dodecameric intasome containing 12 IN subunits. Work done at the same time using Maedi visna virus (MVV) IN (next paragraph), as well as future HIV-1 IN studies [90,128], have indicated that the higher-order forms of lentiviral intasomes are almost certainly biologically relevant.

The initial successes in intasome structural biology with PFV were rooted in the superior biophysical and biochemical properties of the spumaretroviral IN protein [13,21]. Building on this theme, the Cherepanov lab leveraged the IN from the ovine lentivirus MVV to construct lentivirus intasomes. Alison Ballandras-Colas presented the cryo-EM structure of the MVV intasome containing pre-cleaved LTR DNA, which was resolved to 4.9 Å resolution [175]. Eight structurally distinct types of IN subunits overall formed a homo-hexadecamer of MVV IN with a tetramer-of-tetramers architecture. The MVV structure first identified the CIC that is represented by the basic PFV IN tetrameric arrangement and observed in all retroviral intasome structures. The greater-than-tetramer arrangements observed for IN within α-retroviral [176], β-retroviral [14], and lentiviral [90,91,127,175] intasomes is in part due to structural constraints imposed by IN interdomain linker lengths [14]. In such cases, additional IN protomers supply IN CTDs to the CIC structure. The higher-order stoichiometry of IN-to-viral DNA within lentiviral intasomes may also serve to increase the local density of IN-binding partners to enhance the probability of targeting specific genomic loci, e.g., LEDGF/p75-associated genes, for integration [175]. 

#### 3.6.2. Sessions 2A–C: Cellular Regulation of Integration

Although the IN-binding cofactor LEDGF/p75 was known to play an important role in targeting HIV-1 PICs to genes for integration [106,138], knockdown of CA-binding host factors, such as nucleoporin 153, could also disrupt genic integration targeting [177,178]. Two papers published in 2015 seemed to present incongruent models of HIV-1 integration targeting. Marini et al. reported that HIV-1 specifically targeted genes located within the peripheral region of the nucleus for integration [179] while Chin et al. reported that knockdown of the CA-binding host factor cleavage and polyadenylation specificity factor 6 (CPSF6) led to an uncharacteristic buildup of viral complexes at the nuclear periphery [180]. To address the apparent inconsistency, Alan Engelman combined image-based approaches of HIV-1 nuclear location with proviral mapping experiments [181]. In the absence of the CA-CPSF6 interaction, HIV-1 uncharacteristically targeted lamina-associated domains, which interact with peripheral nuclear lamina proteins and are typically heterochromatin, for integration. Although Engelman and colleagues reproduced the result that some favored gene targets of HIV-1 were also peripherally located, the majority were located fairly evenly across cell nuclei. By contrast, genes targeted for integration under conditions of CPSF6 depletion were consistently peripheral [181]. In combination with more recent data, it is thought that the CA-CPSF6 interaction licenses HIV-1 intranuclear penetration for colocalization with nuclear speckles, where integration preferentially occurs in speckle-associated domains of chromatin [182]. 

DNA topology and nuclear compaction, which can significantly impact retroviral/HIV integration frequency [183,184], also play key roles in proviral transcription. Marc Lavigne investigated the effect of DNA topology on the efficiency of HIV-1 integration [184,185]. Modifying topology in infected cells by inhibiting or silencing DNA topoisomerases did not affect the efficiency of integration but did negatively impact viral transcription. DNA topoisomerase I was identified as a potent repressor of HIV-1 transcription that acts by forming a topoisomerase/guanine quadruplex structure in the LTR promoter region. 

Ganjam Kalpana presented an update on the IN-INI interaction and how this influences the late events of virus replication. HIV-1 IN binds INI1/SMARCB1 through the cell factor’s Rpt1 domain. The NMR structure of the INI1-Rpt1 domain was determined, and a molecular model of its interaction with the HIV-1 IN CTD was built [186,187]. Interestingly, INI1-Rpt/CTD interface residues overlap with IN CTD residues required for RNA binding. Moreover, INI1-Rpt1 and HIV-1 trans-activation response (TAR) element RNA competed with each other for IN binding at comparable inhibitory concentrations. The proposed structural mimicry between INI1-Rpt1 and TAR RNA possibly accounts for INI1/SMARCB1’s influence on the late events of HIV-1 replication. 

Pascal Lesage investigated the specificity of LTR retrotransposon Ty1 integration targeting. Ty1 preferentially integrates into 1-kb windows upstream of RNA polymerase III-transcribed genes, with two major sites of integration per nucleosome occurring near the H2A-H2B heterodimer interface. Ty1 IN interacts with different subunits of Pol III, and a short region in Ty1 IN was described as necessary and sufficient for interaction with the Pol III AC40 subunit [188]. Mutations in this region of IN altered Ty1 integration profiles at Pol III-transcribed genes and redistributed Ty1 insertions to chromosome ends. Swapping of the IN-targeting domains between Ty1 and Ty5 enabled Ty5 insertions at Pol III-transcribed genes, highlighting the modular nature of IN-cell factor interactions in Ty-mediated integration targeting. This study further highlighted the types of molecular interactions that have evolved to quell the potential harmful effects of intracellular retrotransposition.

#### 3.6.3. Sessions 3A–C: Integration Selectivity, Gene Therapy, and Latency

Prior studies with HIV-1 and Mo-MLV established cellular LEDGF/p75 and bromodomain and extra-terminal domain (BET) proteins as IN-binding cofactors for targeting their respective PICs to preferred sites of integration (reviewed in ref. [189]). To find potential δ-retroviral IN targeting cofactors, Goedele Maertens performed a mass spectrometry-based screen for cell-binding proteins using HTLV-1 and bovine leukemia virus INs as bait [190]; this study identified the B’ subunit of the heterotrimeric PP2A serine/threonine phosphatase. Purified B’ protein importantly stimulated the concerted integration activities of HTLV-1 and HTLV-2 INs. Despite these promising results, more work is needed to determine the potential role of the IN-B’ interaction in integration targeting during δ-retroviral infection. The comparatively large number of B’ family members and splice isoforms naturally conflates cell-based approaches to testing the IN-B’ interaction.

Dalibor Miklik investigated genomic features permissive for long-term proviral expression, focusing on the epigenetic landscape of integrated α-retroviral vectors [191]. Human myeloid lymphoblastoma K562 cells were transduced with α-retroviral vectors expressing the GFP reporter gene. Integration sites were analyzed in bulk cell (non-selected) populations and in clones selected for GFP expression. Selection led to proviruses being over-represented in transcription units, particularly near promoters. Vectors modified with an anti-silencing CpG island core sequence increased transduction ~10-fold and somewhat increased proviral positionings across genes within selected clones. Overall, these data suggested that integrated proviruses were subjected to gradual epigenetic silencing during long-term cultivation and that promoter/enhancer proximity preferentially guarded proviral gene expression from these cellular silencing effects. 

#### 3.6.4. Sessions 4A–C: Inhibitors, Resistance, and New Therapeutical Approaches

Given the widespread rollout of DTG, it is critical to understand drug resistance mechanisms in great detail. Although resistance to second-generation INSTIs can be generated in vitro, this takes much longer to establish compared to in vitro models of first-generation INSTI resistance; additionally, the de novo IN changes in general confer much lower resistance than do the types of changes that occur when HIV-1 is challenged with first-generation INSTI compounds [192]. Resistance to INSTIs in patients follows similar trends [193]. Unexpectedly, mutations located outside of the IN region of *pol* can confer resistance to DTG [194,195]. Olivier Delelis first described the finding that changes in the 3′ polypurine tract (PPT) can confer significant DTG resistance [194]. Recent research has clarified that retroviral DNA is rapidly heterochromatinized in the nucleus and that certain transcriptional activators, such as HTLV-1 Tax, significantly enhance expression from unintegrated HIV-1 DNA [196]. Studies of DTG resistance that yielded 3′-PPT changes were oftentimes conducted in Tax+ cell lines such as MT4 and C8166, and recent work has clarified that the PPT changes amplify levels of 1-LTR circles to drive replication from the unintegrated DNA templates [197,198]. It is accordingly somewhat unclear to what extent 3′-PPT changes may contribute to clinical DTG resistance, though such changes have been identified in at least one patient [199]. Changes in the HIV-1 *env* gene, which encodes the envelope glycoprotein (Env), can also confer significant resistance to INSTIs such as DTG [200]. In these cases, the changes increased the efficiency of infection mediated by direct cell-cell contact, which seemingly overwhelms the abilities of INSTIs to block the infection. Although Env changes can amass to confer >1000-fold resistance to DTG [201], we are unaware at the time of this writing of data linking such mutations to clinical INSTI resistance. Given the current widespread use of DTG, it is critical to carefully monitor non-canonical pathways to the generation of DTG resistance in addition to direct changes in IN.

## 4. Conclusions and Perspectives

The International Conferences on Retroviral IN/Integration that have taken place since 1995 afford intimate opportunities for like-minded scientists to meet and discuss research results and interests. Outside of these dedicated meetings, integration scientists would generally attend broader-based meetings such as the annual Cold Spring Harbor Meeting on Retroviruses or the annual Conference on Retroviruses and Opportunistic Infections (CROI). Although both are excellent conferences, the larger sizes of these meetings and broader subject matters naturally dilute the integration-tropic experience of the dedicated international meetings that we have described herein. Although 6 years transpired between the first and second IN meetings and then 7 years passed between the second and third conferences, an ad hoc committee that convened at the 2014 Asilomar conference recommended having the meetings every three years, a frequency that was met for the 4th, 5th, and 6th conferences. Due to the omnipresent COVID-19 monkey-wrench, 6 years once again transpired between the 6th and 7th meetings. The community has recommitted to the once-every-three-years frequency, and the 8th International Conference on Retroviral Integration is tentatively scheduled for the summer of 2026.

The combined seven meetings have thus far seesawed between US and European locations, which importantly engages the international community and gives the opportunity to attend at least every other time, given costs and time constraints that can impact international travel. To date, locations have been chosen to match the expertise of the local scientific organizer(s). This model has so far restricted our ability to convene conferences at expanded international locations, e.g., in Asia, South America, or the Middle East. The ad hoc committee that helps to plan the conferences, which includes one of us (Engelman), welcomes feedback from international scientists who may be interested in hosting a future international conference on retroviral integration.

## Figures and Tables

**Figure 1 viruses-16-00604-f001:**
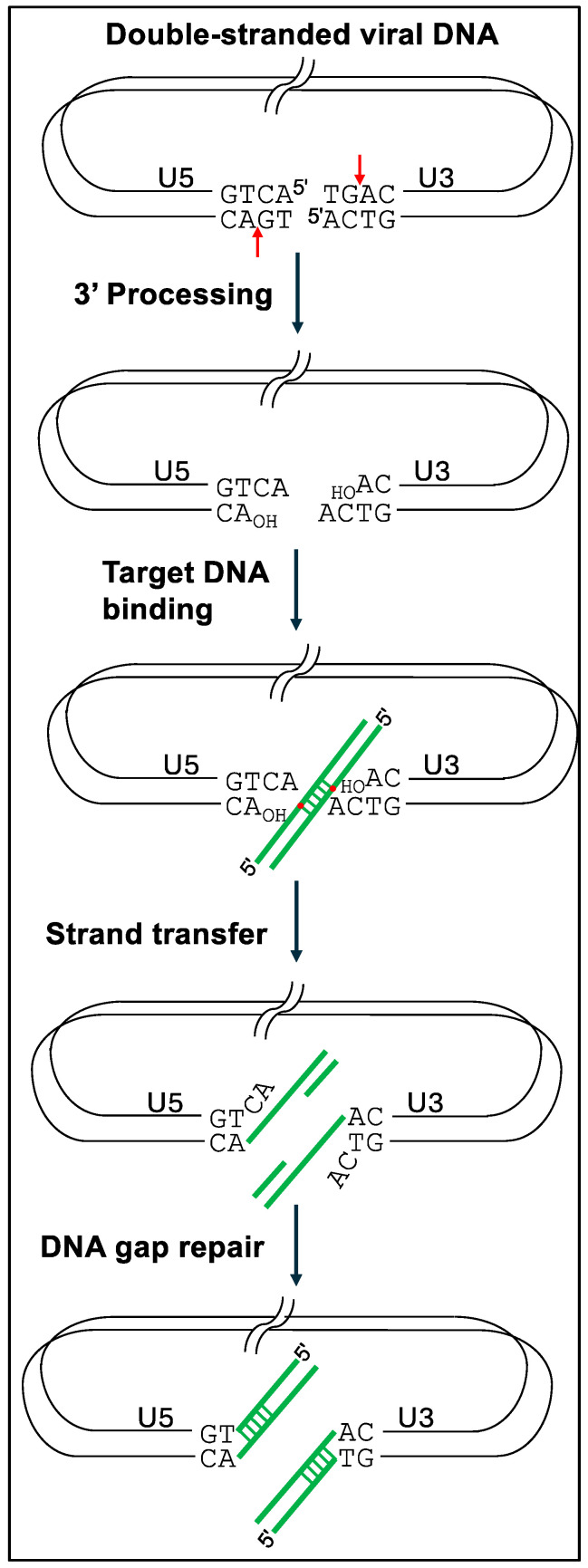
DNA cutting and joining steps of retroviral integration. Depicted is linear HIV-1 DNA (thin black lines) and chromosomal target DNA (thick green lines). Scissile phosphodiester bonds are shown as red vertical arrows (for 3′ processing) and small red circles (for DNA strand transfer). For simplicity, the IN enzyme was omitted from the drawing. See main text for additional details.

**Figure 2 viruses-16-00604-f002:**
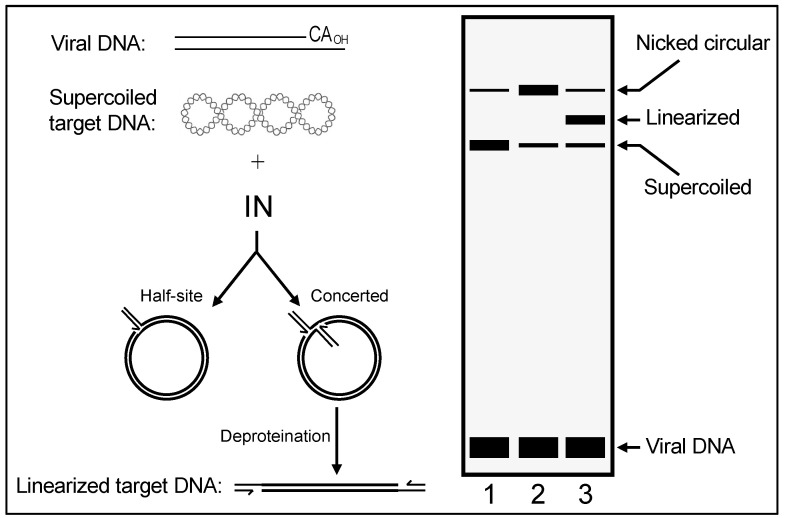
Design of a representative concerted integration activity assay. Shown to the left are a viral DNA oligonucleotide synthesized with a recessed CA_OH_-3′ end and supercoiled target DNA. Supercoiled DNA in particular helps to distinguish products of concerted integration from single viral DNA end integration products, which are referred to as half-site integration. Deproteinized reaction products analyzed by agarose gel electrophoresis, schematized to the right, reveal the positions of unreacted viral and target DNA substrates in the absence of added IN (lane 1). Plasmid DNA isolation techniques invariably nick a fraction of supercoiled molecules, which migrate more slowly through the gel than does the compacted supercoiled population. INs that preferentially catalyze half-site integration, such as HIV-1, yield a predominance of single LTR-tagged circles, which comigrate under these conditions with nicked plasmid circles (lane 2) [12]. Other INs, such as those derived from Moloney murine leukemia virus (Mo-MLV) [3], avian myeloblastosis virus (AMV) [7], prototype foamy virus (PFV) [13], or mouse mammary tumor virus [14], possess robust concerted integration activities, which, after deproteination, yield reaction products that comigrate with linearized plasmid DNA (lane 3). HIV-1 IN concerted integration activity is enhanced through the use of comparatively long viral DNA substrates [9] or the IN-binding host factor lens epithelium-derived growth factor (LEDGF)/p75 [10].

**Figure 3 viruses-16-00604-f003:**
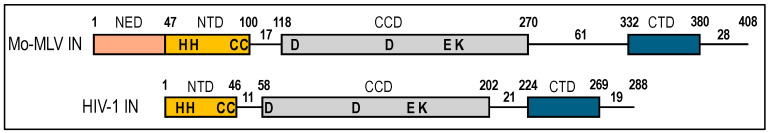
Domain organization of two representative retroviral IN proteins. Conserved amino acid residues highlighted in the main text are shown in single-letter code. The additional lysine (K) in the CCD, which plays a key role in sequence-specific viral DNA binding, is conserved among retroviral INs and some bacterial transposase proteins but is not seen in retrotransposon INs [66]. Numbers refer to amino acid positions of domain boundaries, as well as to interdomain linker and C-terminal tail lengths. NED, N-terminal extension domain present in a subset of retroviral INs (γ-retroviruses, ε-retroviruses, and spumaretroviruses).

**Figure 4 viruses-16-00604-f004:**
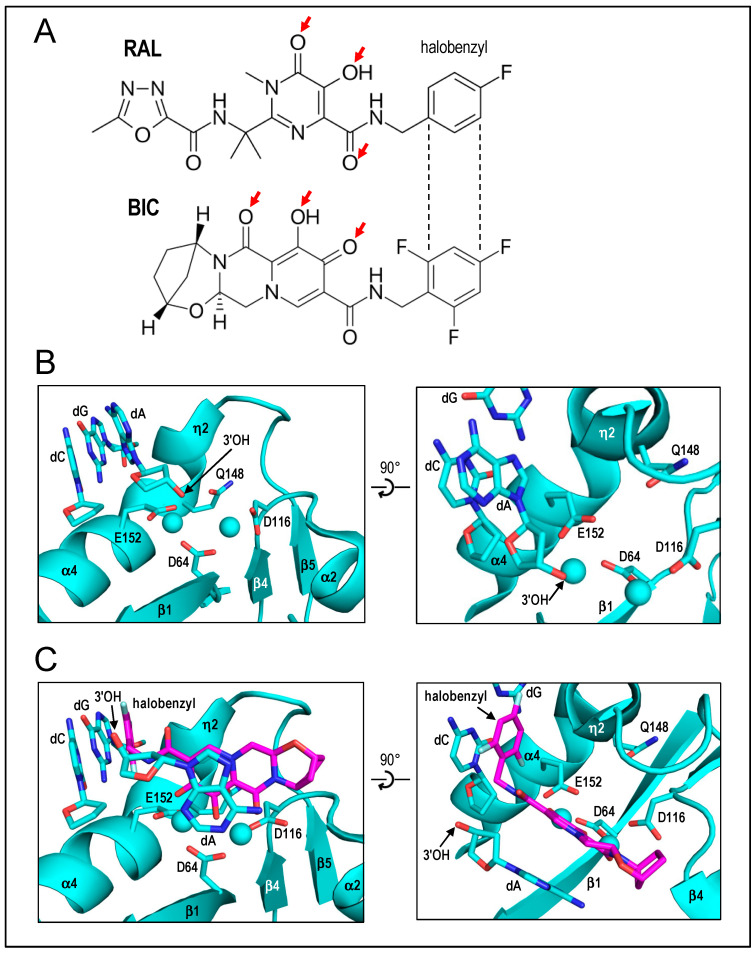
INSTI and intasome structures. (**A**) Structures of representative INSTIs RAL and BIC highlighting conserved heteroatoms (red arrows) and halo-benzyl sidechains. (**B**) Cryo-EM structure of the HIV-1 intasome [Protein Databank (PDB) accession code 6PUT] highlighting DDE catalytic triad residues (D64, D116, E152), CA end of the cleaved viral DNA strand, as well as opposing C-paring dG nucleotide. IN secondary structural elements are labelled. Q148, which can confer significant INSTI resistance when altered, is also highlighted. Spheres, calcium atoms; red and blue, oxygen and nitrogen atoms, respectively. The rightward panel affords an ~90° rotated “top view” of the leftward panel. (**C**) Same as in panel (**B**), except with BIC (magenta backbone with grey fluorines) bound (PDB code 6PUW), which displaces dA along with its 3′-OH required for IN strand transfer activity from committed positions at the IN active site. Spheres, Mg^2+^ ions. Panels (**B**,**C**) based on ref. [90].

**Table 1 viruses-16-00604-t001:** FDA-approved INSTIs and formulations.

Compound	Trade Name	Co-Formulated Compound(s) ^1^	Year of Approval
Raltegravir (RAL)	Isentress	n.a. ^2^	2007
Elvitegravir (EVG)	Stribild	Cobicistat ^3^/FTC/TDF	2012
	Genvoya	Cobicistat/FTC/TAF	2015
Dolutegravir (DTG)	Tivicay	n.a.	2013
	Juluca	RPV	2017
	Dovato	3TC	2019
	Triumeq	ABC/3TC	2022
Bictegravir (BIC)	Biktarvy	FTC/TAF	2018
Cabotegravir (CAB)	Vocabria	n.a.	2021
	Cabenuva ^4^	RPV	2021
	Apretude ^4^	n.a.	2021

^1^ FTC, emtricitabine (NRTI); TDF, tenofovir disoproxil fumarate (NRTI); TAF, tenofovir alafenamide (NRTI); RPV, rilpivirine (non-nucleoside RT inhibitor); 3TC, lamivudine (NRTI); ABC, abacavir (NRTI). ^2^ n.a., not applicable. ^3^ Pharmacokinetic enhancer. ^4^ Extended-release injectable suspension.

## Data Availability

Not applicable.

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
