# Peer review of "Brief Histories of Retroviral Integration Research and Associated International Conferences"

_viruses, 2024, doi:10.3390/v16040604_

Round 1

Reviewer 1 Report

Comments and Suggestions for Authors

This is an extremely interesting and well written article. The retroviral DNA integration field has a very long history and a vast number of papers have been published. This makes it very difficult to follow for researchers who are not actually working on the topic. The approach of taking snapshots of progress through the eyes periodic meetings is novel. It works very well and is quite accessible to general readers interested in leaning more. As someone who has spent most of my career working on retroviral DNA integration it was really interesting to be reminded of the stepping-stones along the path to our current knowledge. It also provides an opportunity to think about old results in the light of what we know now. A lot of work has clearly gene into putting this article together, but the result is well worth the effort.

Minor comments

1.        In the lists talk titles and presenters, the presenters and titles often do not line up. There are also presenters with no talk titles. The instances are too numerous to list, but this needs to be tidied up.

2.        The paragraph at the top of page 3 is somewhat ambiguous. Certainly, concerted integration cannot be identified in reactions that contain only oligonucleotide LTRs. However, in the case of MLV, essentially all integration products are revealed to be concerted under exactly the same reaction conditions when a circular target is included. HIV products are essentially all one-ended under the same conditions used for the oligo assay. Perhaps a distinction between assay design that allows identification of concerted integration products and reaction conditions is needed?

3.        3.5.2. Page 22. Min Li’s talk actually focused on integrase with a short peptide, an AT-hook derived from LEDGF, fused to the N-terminus (P5-IN), not Sso7d IN.

Reviewer 2 Report

Comments and Suggestions for Authors

This is a comprehensive review of the field of retroviral integration over the last 60 years, with particular focus on the first 6 International Conferences on Retroviral Integration.  It is clearly written and an excellent addition to the field.  The Introduction and Early studies sections should be valuable for newcomers to the field.  Thus, this is a useful addition to this Special Issue.

1.  There are 6 large tables, listing all talks presented at the meetings.  These are misaligned, so that the Speaker name does not always align with the beginning of the Title (whenever the title is longer than 1 line).  Also, it would be easier to look at the tables if the Session Titles were integrated into the tables, rather than just shown by numbers with titles in the text.  

2.  Typo:  line 147:  simiispumavirus

Line 277:  much less so able (delete so)

3.  Line 117:  The claim that "the most important consequence of basic scientific research is the ability to inform the development of medicines" is debatable.  I would soften it a bit to "an important consequence...)
